# GenSDF: Two-Stage Learning of Generalizable Signed Distance Functions

**Gene Chou**
Princeton University
gchou@princeton.edu

**Ilya Chugunov**
Princeton University
chugunov@princeton.edu

**Felix Heide**
Princeton University
fheide@princeton.edu

## Abstract

We investigate the generalization capabilities of neural signed distance functions (SDFs) for learning 3D object representations for unseen and unlabeled point clouds. Existing methods can fit SDFs to a handful of object classes and boast fine detail or fast inference speeds, but do not generalize well to unseen shapes. We introduce a two-stage semi-supervised meta-learning approach that transfers shape priors from labeled to unlabeled data to reconstruct unseen object categories. The first stage uses an episodic training scheme to simulate training on unlabeled data and meta-learns initial shape priors. The second stage then introduces unlabeled data with *disjoint classes* in a semi-supervised scheme to diversify these priors and achieve generalization. We assess our method on both synthetic data and real collected point clouds. Experimental results and analysis validate that our approach outperforms existing neural SDF methods and is capable of robust zero-shot inference on 100+ unseen classes. Code can be found here.

## 1 Introduction

Learned 3D representations form the core of shape recovery [1], robotic manipulation [2], scene understanding [3], and content generation [4] tasks. While simple to interpret, high-quality explicit 3D representations can come with a slew of hidden costs. Conventional voxel grids without acceleration suffer from cubical memory requirements [5, 6, 7], dense point clouds contain large amounts of redundant information make querying local structures computationally expensive [8, 9], and direct optimization of mesh representations is often infeasible [10]. As an alternative, works such as [11, 12, 13] propose *implicit* learned representations; networks trained to functionally approximate some underlying 3D geometry. These methods can offer lower memory requirements and faster training and inference speeds [14] than conventional explicit representation methods [5, 8].

Neural signed distance functions (SDFs) [11] are a subset of this work which implicitly model the distance from a queried location to the nearest point on a shape's surface – negative inside the shape, positive outside, zero at the surface. Existing methods are capable of approximating diverse synthetic geometry, including multiple objects [11, 15, 16] and varying levels of detail for single objects [17, 18]. However, most successful methods are fully supervised–they require access to ground truth signed distance values to fit to an object. As ordinary point clouds lack these values, a fully-supervised approach cannot learn from such unlabeled data.

To replace the need for any annotated data, fully unsupervised approaches [19, 20, 21] operate directly on point clouds. These methods leverage pseudo-labels and proxies for ground truth signed distance values, for example the unsigned distance to the nearest point in the point cloud. Unfortunately, existing unsupervised methods can only reconstruct a handful of shapes, and break down when we introduce just three or four additional additional object categories. As a result, these approaches can also not take advantage of large unlabeled point cloud datasets [22, 23, 24] and their real-world applications are limited. Our proposed method lifts this limitation, and learns from large-scale

36th Conference on Neural Information Processing Systems (NeurIPS 2022).

unlabeled data, successfully reconstructing an order of magnitude more categories in inference than seen during training.

To achieve this we propose a semi-supervised learning approach. Existing semi-supervised methods use labeled and unlabeled data of the same class to improve intra-class performance for tasks such as 3D reconstruction from images [25, 26, 27] and scene reconstruction from point clouds [3, 28]. Although they perform well on seen data, these methods are trained on labeled and unlabeled data of the same class, and so do not by themselves generalize well to unseen classes. We propose a novel semi-supervised meta-learning learning approach that learns geometric relations between labeled and unlabeled data of non-overlapping categories. One which learns an SDF that generalizes to a diverse set of unseen, out-of-distribution geometries.

We train in two stages. In the first meta-learning stage, we learn shape priors from the labeled set. Different from existing supervised methods, we introduce an episodic training scheme where we split data into subsets with disjoint classes to learn a generalized shape representation. One part of this split is treated as a labeled supervised set of data, with ground truth signed distance penalties, while the other is used to emulate unlabeled data samples, with only self-supervised losses. We validate that this first stage provides a generic shape prior which favors expressiveness at the cost of reconstruction quality. In the second, semi-supervised stage, we introduce a new fully disjoint set of unlabeled data to the training scheme. Successful training hinges on our novel self-supervised sign prediction loss, which diverges from the unsigned convention of previous works [19, 20, 21]. This second stage allows the model to ingest large amounts of unlabeled data to diversify the shape model learned in the first stage, which significantly increases the quality of the signed distance predictions over a wide set of unseen shape classes.

We evaluate our method on Acronym [29], a synthetic dataset, and YCB [23], a collection of real-world point clouds. We validate the generalization capabilities of our approach, and find that it outperforms state-of-the-art learned SDF methods and is capable of robust zero-shot inference [30] on 100+ unseen classes and scanned point clouds.

## 2 Related Work

**Learning Implicit Surface Representations.** Implicit neural representations form the backbone of many modern 3D reconstruction and view synthesis works [11, 12, 13, 31, 32]. They use neural networks as *universal approximators* [33] to directly learn a mapping between spatial or spatio-temporal [34] coordinates and scene attributes, which offers a fully-differentiable and space-efficient [28] way to represent 3D geometry. Park et al. [11] first experimented with mapping coordinates to signed distance values – a neural signed distance function (SDF) – whereby an object is implicitly represented by the distances between 3D coordinate queries their closest surface point.

With labeled data – when all 3D query points are annotated with ground truth signed distance values – predictions of signed distance values can be supervised directly [11, 15]. Existing methods [11, 15] require test-time fine-tuning even for a single class, prohibiting real-time inference or prediction on unlabeled data, or overfit to single objects [17, 18]. Occupancy-based methods [16, 35, 36] can learn a rudimentary shape prior, but still struggle to reconstruct fine features in unseen shapes. The proposed semi-supervised method learns a robust shape prior, and substantially outperforms fully supervised methods when tested on a set of over one hundred unseen categories.

With only unlabeled data – un-annotated point clouds – approaches try to generate proxies for signed distance values for supervision. Ma et al. [19] look at the unsigned distance to the nearest point in the point cloud as a proxy, and Atzmon and Lipman [20] propose a class of loss functions and geometric initializations such that these unsigned distances converge to signed values. We propose a signed nearest neighbor loss and demonstrate that our approach, unlike existing unsupervised methods, is able to reconstruct thin object structures and converge on a dataset with large object variance.

**Semi-supervised Surface Representations.** Traditional semi-supervised methods rely on learning from datasets where each category contains labeled and unlabeled samples. During inference they rely on pseudo-labels [37, 38], or employ ensemble techniques to enforce consistency [39]. These methods cannot be directly applied to target data from unseen categories.

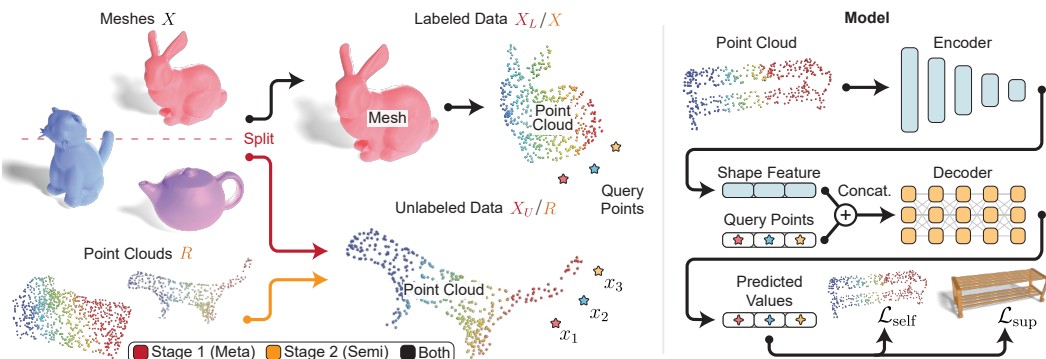

Figure 1: Our two-stage learning pipeline. The first stage (**red**) episodically splits the dataset into two subsets of disjoint categories of emulated labeled and unlabeled data, trained with combined supervised $\mathcal{L}_{\text{sup}}$ and self-supervised $\mathcal{L}_{\text{self}}$ loss. The second stage (**orange**) introduces real unlabeled data with non-overlapping classes in a semi-supervised fashion to improve model generalization.

For 3D representation tasks, existing methods [25, 26, 27] use semi-supervision to reconstruct geometries from images or point clouds [3, 28]. Elich et al. [25] pretrain their model with labeled data to learn 3D representations, to recover known shapes from RGBD data. Li et al. [3] use labeled data as dense boundary values for semi-supervised SDF learning, and fit road scenes from point clouds. While these approaches improve predictions on seen data, they cannot generalize well to unseen classes, even with test-time optimization [3]. Recently, person re-identification and domain adaptation methods perform semi-supervision on non-overlapping labeled and unlabeled classes and have shown to generalize better [40, 41, 42]. This has not been attempted for learning SDFs, and our method introduces a meta-learning approach to learn shape priors, where we emulate training on unlabeled data with pseudo-labels.

**Generalization.** The ability to learn generalizable 3D representations is necessary for real-world applications, as real objects exhibit significantly more diversity than any dataset. In learning implicit functions, one approach [43, 44] is to break down shape features to find common properties, but this also requires prior knowledge about target shapes which might not be available.

We propose a meta-learning approach that does not make assumptions or require prior information about target data. Meta-learning has been extensively studied in few-shot learning [45, 46], domain generalization [47], and image retrieval [39], 3D reconstruction from 2D or 2.5D images [48, 49, 50], and point cloud classification and segmentation tasks [51, 52, 53]. However, meta-learning has not been extensively investigated in the context of SDFs. The most relevant method is MetaSDF [15], albeit it is fully supervised and exhibits large time and memory complexity due to inner loop gradient descent steps. Furthermore, MetaSDF overfits to single objects and requires ground truth signed distance values during test time. The proposed episodic training approach avoids this costly inner loop, and learns a unified network that only requires a raw input point cloud for reconstruction.

## 3 Semi-supervised Meta-Learning of Implicit Functions

We next formalize the problem we are solving in Sec. 3.1, then describe the proposed learning method and training objectives in Sections 3.2 and 3.3.

### 3.1 Problem Formulation

**Conditional Signed Distance Functions** We are interested in learning a generalized representation for shapes. We learn a signed distance function (SDF), which is a continuous function that represents the surface of a shape approximated by the zero level-set of a neural network. Given a raw input point cloud $P = \{p_i \in \mathbb{R}^3\}_{i=1}^N$ with $N$ points and a 3D query point $x \in \mathbb{R}^3$, we learn a conditional SDF $\Phi : \mathbb{R}^3 \times \{p_i \in \mathbb{R}^3\}_{i=1}^N \to \mathbb{R}$ such that the function $\Phi$ predicts the signed distance value for a 3D coordinate conditioned on a sparse point cloud. Then $\Phi$ is a unified function that can be directly applied to any target point cloud.

$$x, P \mapsto \Phi(x, P) = s, \tag{1}$$

where $s$ denotes the predicted signed distance value between $x$ and the shape described by $P$.

The surface boundary of a shape described by $P$ is its zero-level set $S_0(\Phi(P))$, which can be expressed as

$$S_0(\Phi(P)) = \{z \in \mathbb{R}^3 \mid \Phi(z, P) = 0\}. \tag{2}$$

Thus, given a trained $\Phi$, we can visualize the surface of some $P$ by drawing its zero-level set.

**Problem Setting**  We assume both labeled and unlabeled datasets are available, denoted $X$ and $R$, respectively. $X = \{P_X, \text{SDF}\}$, where $P_X$ is a set of point clouds and $\text{SDF}(\cdot)$ denotes the ground-truth SDF operator that is defined for all query points $x \in \mathbb{R}^3$. $R = \{P_R\}$, where $P_R$ is a set of point clouds belonging to classes disjoint with those of $P_X$. Each point cloud is a set of 3D coordinates that describes the surface boundary of an object, and contains no additional information such as normals.

To learn a generalizable conditional SDF, we propose a two-stage semi-supervised meta-learning algorithm. In the first stage, we aim to learn shape priors using $X$. As illustrated in Figure 1, we propose to train our model using an episodic scheme [47], where every $f$ epochs we randomly split $X$ into two subsets with *disjoint* categories. In the second stage, as shown in Figure 1, we continue to train the pretrained weights from the first stage and train the model on $X$ and $R$ in a semi-supervised fashion. We obtain a unified neural network $\Phi$ that learns from one dataset and can generalize to represent SDFs of objects in unseen categories during test time; i.e., zero-shot inference [30].

### 3.2 Stage 1: Meta-Learning a Shape Prior

Our eventual goal is to train on labeled and unlabeled data simultaneously, so we mimic this goal in a meta-learning approach as we learn shape priors from $X$. Every $f$ epochs, we split $X$ into two subsets $X_L$ and $X_U$ with disjoint categories. We train $X_L$ with a supervised loss $\mathcal{L}_{\text{sup}}$, similar to typical supervised approaches on labeled data. We take away ground-truth signed distance values from $X_U$ making it "pseudo-unlabeled", and train it with our self-supervised loss $\mathcal{L}_{\text{self}}$. Our training objective is to minimize

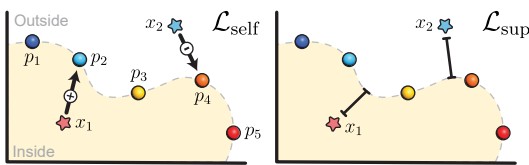

Figure 2: Our self-supervised (**left**) loss uses the nearest neighbor of a query point to approximate *signed* distance. The supervised loss (**right**) instead penalizes based on exact signed distance.

$$\mathcal{L}_{\text{meta}} = \mathcal{L}_{\text{sup}}(X_L) + \lambda_m \mathcal{L}_{\text{self}}(X_U) \tag{3}$$

where $\lambda_m$ controls the strength of the self supervised loss $\mathcal{L}_{\text{self}}$. This meta-learning approach intends to accomplish two goals. First, the model learns to backpropagate with our self-supervised loss, which we need for training on unlabeled data in the second stage. Second, episodic training on non-overlapping categories forces the model to learn a generalized shape representation rather than overfit to geometries within a single category.

Compared to directly training on labeled and unlabeled data, this meta-learning process provides flexibility as we can tune the split frequency and ratio of labeled and "pseudo-unlabeled" data. The model repeatedly sees the same point clouds with and without labels, so shape features generated during labeled splits are shared with "pseudo-unlabeled" splits. In contrast, we find that semi-supervised training from scratch is unstable because errors in unlabeled data accumulate (see Tab. 3). We empirically show in our ablation study that our method has stronger representation power for both seen and unseen classes. We provide an example of a training step in Alg. 1 and a diagram in Fig. 1. Next, we explain our loss functions in more detail.

**Supervised Loss Component**  In the supervised setting, we have the ground truth SDF operator; i.e., we know the signed distance values for all query points $x$. Therefore, the loss function is simply

$$\mathcal{L}_{\text{sup}} = \frac{1}{K} \sum_{k \in K} \|\Phi(x_k) - \text{SDF}(x_k)\|_1, \tag{4}$$

where $\| \cdot \|_1$ is the L1 loss.

---

**Algorithm 1** Stage 1: Meta-Learning a Shape Prior

---

1: **preprocess**: a labeled set $X$
2: **initialize**: learning rate $\eta$, split frequency $f$, hyperparameter $\lambda_m$, model $\Phi$ with parameters $\theta$
3: **for** $e$ **in range**(num_epochs):
4:     **if** $(e \bmod f) == 0$:
5:         $X_L, X_U \leftarrow \texttt{split}(X)$
6:         $\mathcal{L}_{\mathrm{sup}} = \frac{1}{|X_L|} \sum\limits_{(x,\mathrm{SDF}) \in X_L} \|\Phi(x;\theta^e), \mathrm{SDF}(x)\|_1$            ▷ Eq. (4)
7:         $\mathcal{L}_{\mathrm{self}} = \frac{1}{|X_U|} \sum\limits_{(x,t) \in X_U} \|(x \pm \frac{x-t}{\|x-t\|} \cdot \Phi(x;\theta^e)), \ t\|_2^2$     ▷ Eq. (7), simplified
8:         $\mathcal{L}_{\mathrm{meta}} = \mathcal{L}_{\mathrm{sup}} + \lambda_m \mathcal{L}_{\mathrm{self}}$                     ▷ Eq. (3)
9:         $\theta^{e+1} \leftarrow \theta^e - \eta \nabla \mathcal{L}_{\mathrm{meta}}$
10: **return** $\theta$

---

**Self-supervised Loss Component for Unlabeled Training Samples**     For each point $x \in \mathbb{R}^3$, we use its closest point $p \in P$ to approximate its projection to the surface, defined as

$$t = \operatorname*{argmin}_{p \in P} \|x - p\|_2. \tag{5}$$

In the literature [54], it is common to use

$$\hat{t} = x - \nabla \Phi(x) \Phi(x) \tag{6}$$

to approximate the closest point $t$ on the object surface to query point $x$ given some trained SDF function $\Phi$ for tasks such as collision detection. We build on this insight: we want to predict $\hat{t}$ to approximate $t$ on the point cloud by using the predicted signed distance value. However, we find that directly using Eq. (6) leads to highly inaccurate sign predictions. There are no explicit penalties for predicting the wrong sign and when the query point is close to the surface, an $\ell_2$ loss between $\hat{t}$ and $t$ can be very low even when the sign is predicted incorrectly. For a surface boundary to be accurately estimated, there must be accurate sign changes between 3D coordinates.

Thus, we substitute $\nabla \Phi(x)$ with the direction vector between $t$ and $x$. For incorrect sign predictions, the vector points in the direction opposite to the surface, guaranteeing a larger error. We denote our approach self-supervised because the predictions of the signs are used as labels. This makes it possible to leverage ground-truth signs during the meta-learning stage to guide training since the sign and absolute distance can be disentangled. In contrast, $\nabla \Phi(x) \Phi(x)$ is calculated as a unit that points in the direction of maximum distance change. Our training objective is to minimize the distance between $\hat{t}$ and $t$. Furthermore, we add an additional $\ell_1$ loss by using the points on point clouds as they have a distance value of 0, which helps with thinner structures. Our final loss function is

$$\mathcal{L}_{\mathrm{self}} = \frac{1}{K} \sum_{k \in K} \|\hat{t}_k - t_k\|_2^2 + \lambda_p \frac{1}{N} \sum_{p \in P_{\mathrm{unlab}}} \|\Phi(p_n)\|_1, \tag{7}$$

where $\lambda_p$ determines the weight of the point cloud predictions and

$$\hat{t} = \begin{cases} x - \dfrac{x-t}{\|x-t\|} \ \Phi(x) & \text{if } \Phi(x) \geq 0, \\[3mm] x + \dfrac{x-t}{\|x-t\|} \ \Phi(x) & \text{if } \Phi(x) < 0. \end{cases} \tag{8}$$

Previous work [19] has predicted $t$ by directly calculating Eq. (6) during training, but their resulting sign predictions are substantially less accurate than ours for the reasons we discuss. We validate this with the results shown in Tab. 1. We note that a point cloud $P$ is sampled from a surface and does not represent the continuous boundary. Thus, $t$ is the nearest neighbor that exists in $P$ but not necessarily the true one that exists in the target surface we want to reconstruct. As a result there are inaccuracies and previous unsupervised work [19] that use $P$ as a proxy are not robust when training on multiple classes. Our two-stage approach lifts this constraint.

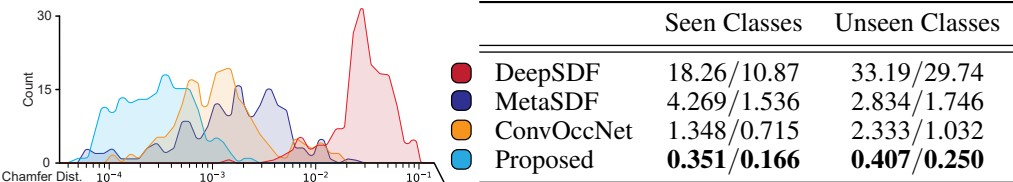

Figure 3: Mean/median Chamfer Distance (CD) of 20 seen and 166 unseen classes (**right**, scale is $10^{-4}$). For our method we report numbers prior to any test-time refinement. We include a histogram plot for visualizing the per-class CD distribution of the unseen classes (**left**, log scale).

## 3.3 Stage 2: Semi-supervision with Unlabeled Training Samples

In our second stage, visualized in Fig. 1, we train our conditional SDF, pretrained in the meta-learning stage, on both labeled and unlabeled data in a semi-supervised fashion. As a result of the learned priors, training remains robust even with large amounts of unlabeled data. Furthermore, continuing to train with labeled data provides strong supervision to ensure inaccuracies from unlabeled data do not build up. We show in our ablation study (Sec. 4.4) that semi-supervised training from scratch can be unstable. Given labeled set $X$ and unlabeled set $R$, our training objective is to minimize

$$\mathcal{L}_{\text{semi}} = \mathcal{L}_{\text{sup}}(X) + \lambda_s \mathcal{L}_{\text{self}}(R) \tag{9}$$

where $\lambda_s$ controls the strength of the self supervised loss $\mathcal{L}_{\text{self}}$.

## 4 Experiments

In this section, we analyze and validate the proposed representation method for seen and unseen object categories. We begin with a discussion of implementation details and experimental setup.

**Implementation** For our experiments, we borrow the encoder architecture from [16] which is a modified PointNet [8] that uses plane projection to learn local geometric features, and parallel UNets [55] to aggregate spatial information. For the decoder we use an 8-layer multi-layer perceptron with 512-dimensional hidden layers, similar to the architecture from [11, 15, 19, 20]. We emphasize, that our two-stage training pipeline is designed to be architecture-agnostic, and we provide results for alternate encoders and decoders in the supplement. For Eqs. (3) and (9) we the set the loss coefficients to $\lambda_m = \lambda_s = 0.1$, and $\lambda_p = 0.01$ in Eq. (7). We provide a full architecture description and additional training details in the supplemental document.

**Experimental Setup** We source training and evaluation data from Acronym [29] and YCB [23]. Acronym is a post-processed subset of the popular ShapeNet [56] dataset which contains 8,872 clean, synthetic 3D models in 262 shape categories. In contrast to ShapeNet, all of Acronym's meshes are *watertight*, which ensures that they have well-defined SDFs. We use the 20 largest classes (2,995 meshes) as a labeled dataset $X$ for all fully-supervised methods, including our meta-learning first stage. The next 76 classes (3,408 meshes) form our unlabeled dataset $R$ for semi-supervised training. The remaining 166 classes (1,525 meshes) are withheld for testing. Preprocessing details can be found in the supplemental document. The YCB dataset [23] contains point clouds from 3D scans of common real-world objects and does not contain ground truth meshes. We use these point clouds as additional unlabeled test data to assess SDF predictions on out-of-distribution objects.

**Evaluation** Given a point cloud $P$ that describes the surface of an object, the proposed method returns an implicit signed distance function $\Phi(P)$. We construct a cube with each dimension ranging from -1 to 1, then discretize it into a set of grids with resolution $256^3$. We run Marching Cubes [57] at each grid point to reconstruct a mesh surface for qualitative assessment. For quantitative evaluation we measure the Chamfer Distance [8] between 30,000 randomly sampled points on the surfaces of the reconstructed and ground truth meshes.

**Experiments** We conduct experiments to investigate the generalizability and representational power of our method. Specifically, we analyze the effects of reconstructing seen and unseen objects (Sec. 4.1), self-supervision on unlabeled data (Sec. 4.2), performance on real-world point cloud scans (Sec. 4.3), and ablation experiments (Sec. 4.4).

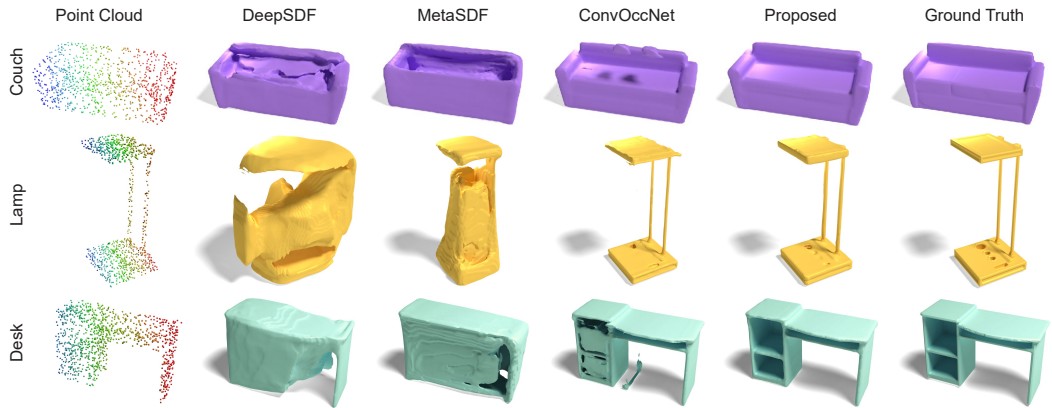

Figure 4: Reconstruction results for seen objects – objects in the training set. DeepSDF and MetaSDF struggle to recover thin structures and non-convex surfaces due to their lack of learned shape priors. Compared to ConvOccNet our approach produces fewer artifacts in empty space.

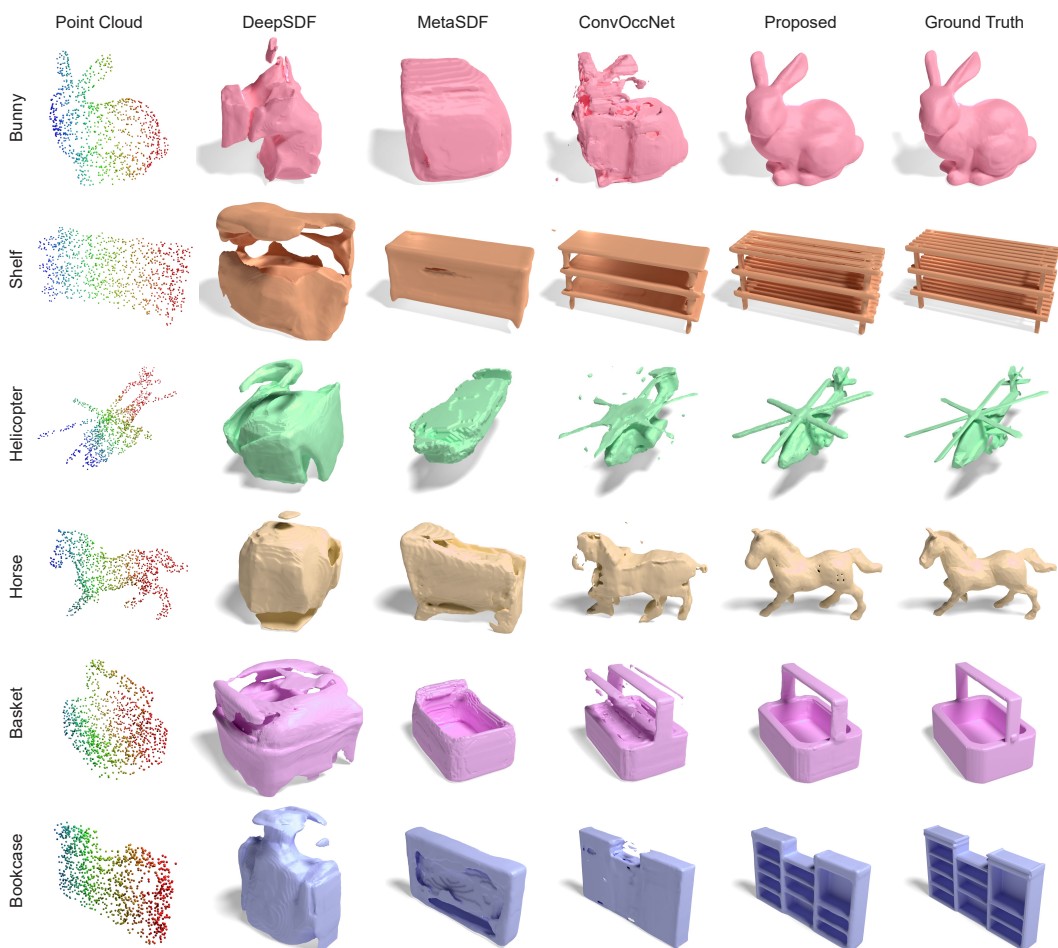

Figure 5: Reconstruction results for *unseen categories* – not included during training. Only our method recovers fine detail, such as the gaps in *Shelf*'s surfaces and the curves of *Bunny* and *Horse*.

## 4.1 Reconstructing Seen and Unseen Shapes

In this subsection, we compare our method to existing approaches: DeepSDF [11], MetaSDF [15], and Convolutional Occupancy Networks (ConvOccNet) [16] on both seen and unseen shapes.[1] For reconstructing seen data, we evaluate results on the 20 shared classes (see the experimental setup paragraph in Sec. 4). For testing on unseen classes, we use the remaining 166 categories that are *disjoint* with any class used for training. To improve visual results, we apply refinement by fitting the model on the point cloud for a few iterations. Specifically, we employ our self-supervised loss without using any additional information beyond 5,000 raw input points. In contrast, DeepSDF [11] and MetaSDF [15] use ground truth signed distance values for refinement. We include specific refinement details in our supplement. Since ConvOccNet [16] does not perform refinement, we report quantitative results of our proposed method *without* refinement for a fair comparison.

**Seen Categories.** Fig. 4 shows reconstruction results on seen shapes. We observe that existing methods and our approach are capable of reconstructing seen data. However, DeepSDF and MetaSDF fail to reconstruct thin structures such as the lamp poles due to the lack of a shape prior. Quantitatively, reported in the middle column of Fig. 3, even though we use the same encoder as ConvOccNet [16] to learn shape priors, our two-stage approach achieves a lower CD by an order of magnitude.

**Unseen Categories.** When tested on unseen categories, as shown in Fig. 5, only our proposed method is able to reconstruct unseen categories, and accurately represents fine details such as narrow gaps in the Shelf example. Quantitatively, as shown in the right column of the table in Fig. 3, the proposed method achieves better performance. Notably, the gap between the CD of seen and unseen classes is lower in our case. While the CD of most methods double between seen and unseen classes, there is only a marginal increase for ours.

## 4.2 Comparison to Unsupervised Methods on Unlabeled Data

A self/unsupervised method is essential for our semi-supervised stage to train on unlabeled data and for test-time refinement on only the raw input point cloud. In this subsection, we compare to existing unsupervised baselines: NeuralPull [19] and SAL [20]. For a fair comparison, we use only our self-supervised loss in Eq. (7) to train directly on unlabeled data and do not perform test-time refinement (*Proposed (self)* in Tab. 1).

Tab. 1 validates that when training a single Queen Bed class, our explicit sign penalty allows our self-supervised method to reconstruct sharper details from irregular surfaces. On the

Table 1: Mean/median Chamfer Distance for self- and un-supervised methods when trained on 5 objects from the same *QueenBed* class and 352 objects from 4 classes (*QueenBed*, *Plant*, *TrashBin*, *DeskLamp*). We report pre-refinement numbers for our approach, scale is $10^{-4}$.

|  | Queen Beds | Four Classes |
| --- | --- | --- |
| SAL | 1.786 / 1.869 | 65.55 / 49.38 |
| NeuralPull | 1.745 / 1.502 | 2.419 / 1.812 |
| Proposed (self) | 0.146 / 0.139 | 0.218 / 0.184 |
| Proposed | **0.081** / **0.077** | **0.089** / **0.061** |

other hand, existing methods [20, 19] produce low losses for close approximations to the absolute values of the signed distances, rather than accurate sign predictions, resulting in loss of details where there is abrupt change in the surface (see supplement for visualizations). All non-supervised methods we test on, including ours, degrade when training degrade quickly as the number of meshes increases.

The proposed method with semi-supervision scales well as we increase the number of classes. Thus, our method can tackle large-scale operations and take advantage of the availability of large amounts of unlabeled data. We also show in our supplement that using our self-supervised variant from this section leads to significantly more robust training.

## 4.3 Reconstruction of Real-World Raw Point Clouds

Next, we test our model on the YCB [23] dataset, which is a real-world point cloud dataset acquired from multi-view RGBD captures. The fused multi-view point clouds in this dataset resemble

---

[1]Approaches such as [17, 58, 18] are designed to overfit to single objects, which significantly deviates from the focus of this paper, so we do not include comparisons with these methods.

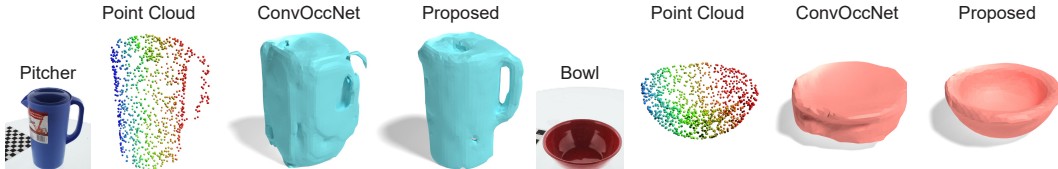

Figure 6: Reconstruction results for real scanned data from unseen categories in YCB [23]. This is a difficult setting, as the data has a different canonicalization, scaling, and distribution. Our proposed method is able to recover structures such as *Pitcher*'s handle and the inside of *Bowl*.

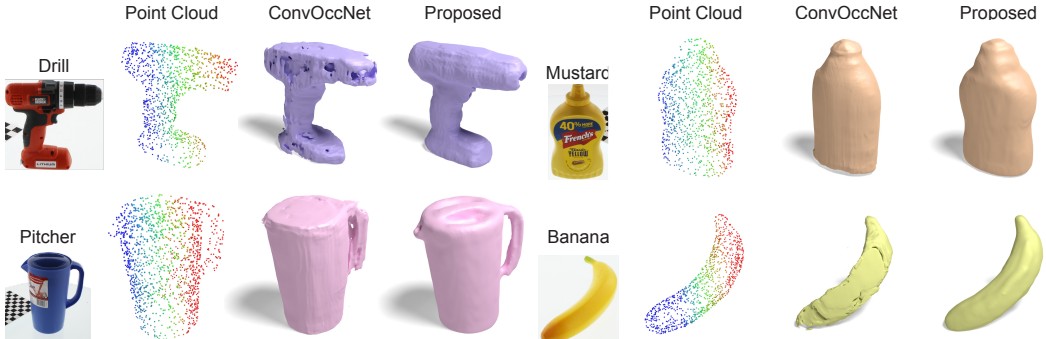

Figure 7: Reconstructions of YCB [23] after recentering and normalization of input point cloud.

input measurements for a robotic part-picking or manipulation task. We demonstrate robust mesh reconstructions of the measured data, e.g., recovering the "handle" of a pitcher in Fig. 6 and 6, which may serve as input to complex robotic grasping tasks.

YCB is a different domain from Acronym [29] and we show two experiments to thoroughly test performance of out-of-distribution data. First, we perform no scaling or normalization of the input point cloud, shown in Fig. 6. Then, we also recenter and normalize the input point cloud (which we would do in a real-world setting) and show results in Fig. 7. We omit results from DeepSDF [11] and MetaSDF [15] since they require ground truth signed distance values during test time. As shown in Fig. 6 and 7, ConvOccNet [16] loses detail such as the handle of the pitcher. Our proposed method is able to generate 3D shapes with detail and plausible geometry, even generalizing to different scaling, illustrating its ability of handling in-the-wild and out-of-distribution data.

To further understand what level of noise our method is susceptible to, we gradually add Gaussian noise with mean zero and variance $\sigma^2$ to the *Pitcher* point cloud in YCB [23] and evaluate its CD. Both quantitatively and qualitatively, our method is able to adapt to noise with $\sigma^2 < 0.5$, but afterward the reconstruct object degrades severely. We show results in Table 2. In the future, adding more diverse unlabeled data to the training of the proposed model, including different domains, scaling, and noise, may further boost the generalizability to any target data.

Table 2: Adding Gaussian noise with zero mean and variance $\sigma^2$ to the *Pitcher* point cloud in YCB and evaluating Chamfer Distance (CD), scale is $10^{-4}$.

| $\sigma^2$ | 0.0 | 0.01 | 0.05 | 0.1 | 0.15 | 0.2 |
|---|---|---|---|---|---|---|
| CD | 0.870 | 0.911 | 3.122 | 5.208 | 7.429 | 7.391 |

## 4.4 Ablation Experiments

In this subsection, we analyze individual components of our two-stage learning. We show quantitative results in Tab. 3. *Only meta* refers here to the meta-learning first stage on 20 classes. This outperforms existing fully supervised methods (see Fig. 3) for both seen and unseen data, while using the same amount of data and no additional manual annotations.

The favorable results validate the generalization capabilities of our episodic training scheme. Our semi-supervision stage further improves reconstruction quality by introducing more shapes without requiring annotations. *Only semi* refers to only training the proposed semi-supervision stage from scratch on all classes without transferring the learned representation from the meta-learning stage. Even though this stage has access to both labeled and unlabeled data, it has a higher CD compared to our final two-stage approach. This is because errors in the proxies of unlabeled data accumulate and result in training that is less robust. This indicates that our meta-learning stage indeed builds a strong foundational prior that ensures the second stage semi-supervised training is robust.

Table 3: Mean/median Chamfer Distance (CD) for various ablations on 20 classes of seen data and 166 classes of unseen data. We report pre-refinement numbers, scale is $10^{-4}$.

|  | Seen Classes | Unseen Classes |
| --- | --- | --- |
| Proposed | **0.351/0.166** | **0.407/0.250** |
| Only meta | 0.503/0.218 | 0.515/0.297 |
| Only semi | 0.821/0.510 | 0.889/0.612 |
| Sup. 20 cls | 0.695/0.468 | 1.164/1.367 |
| Sup. 76 cls | 0.713/0.550 | 0.908/0.866 |

*Sup. (20 cls)* refers to training on the labeled set of 20 classes without episodic training and resplitting. *Sup. (76 cls)* refers to training on 76 classes with ground truth annotations but without episodic training and resplitting. The CD for seen classes is similar to the *Only meta* setting, but our meta-learning method outperforms both baselines by a large margin when it comes to reconstructing unseen classes, despite being trained with fewer categories.

## 5   Conclusion

In this work, we devise a two-stage semi-supervised meta-learning approach that achieves strong generalization to unseen object categories. This method is able to fit to any target point cloud without additional annotations and outperforms existing SDF methods in both seen and unseen categories. Our method is not without limitations, which we plan to address in the future. Similar to existing SDF representations [16], we do not achieve real-time inference due to large number of model parameters. Learning from large-scale multi-object point clouds, with abundant datasets for automotive scenes [59, 60] is also an exciting future direction. The proposed approach may also serve as a representation for inverse rendering methods that tackle vision tasks in an analysis-by-synthesis approach.

## Acknowledgments

We would like to thank Hong-Yuan Liao and Chien-Yao Wang from Academia Sinica, Taiwan for constructive discussions and advice in completing this work. Ilya Chugunov was supported by an NSF Graduate Research Fellowship. Felix Heide was supported by an NSF CAREER Award (2047359), a Sony Young Faculty Award, a Project X Innovation Award, and an Amazon Science Research Award.

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
