# GenSDF
# Supplementary Information

**Gene Chou**
Princeton University
gchou@princeton.edu

**Ilya Chugunov**
Princeton University
chugunov@princeton.edu

**Felix Heide**
Princeton University
fheide@princeton.edu

## Contents

## 1 Implementation Details

In this section, we provide further details for reproducibility, including data preprocessing, training steps and hyperparameters, and model architecture. We will also release all code to facilitate experiments.

### 1.1 Data Processing

**Preprocessing** We recenter and normalize all objects in the Acronym [1] dataset. We use the normalized object coordinate system (NOCS) [2]. This means we contain all points in 3D space in a cube and uniformly scale each object such that the diagonal of its tight bounding box has a length of 1 and is centered within the cube. Different from the original definition [2] where the cube

36th Conference on Neural Information Processing Systems (NeurIPS 2022).

has coordinates ranging from (0,0,0) to (1,1,1), we set our cube to have coordinates ranging from (-1,-1,-1) to (1,1,1). That is, the length of each side on our cube is 2 instead of 1.

We do not perform this preprocessing for the YCB [3] dataset which we use for testing. The point clouds are obtained from real-world scans and have size-accurate scaling. This makes testing difficult since the scaling is out-of-distribution.

**Labeled Data**    For labeled data, we largely follow [4] for preprocessing. From each mesh we sample 235,000 points on the surface. These points form a point cloud $P = \{p_i \in \mathbb{R}^3\}_{i=1}^{235000}$ that describe the surface and have signed distances of 0. Then, for each point $p_i$, we sample two Gaussian distributions with mean 0 and standard deviation 0.005 and 0.0005, respectively. We add these distributions to $p_i$ to obtain two query points per surface point. Since we have the mesh, we can calculate ground truth signed distance values for supervised training and we store all points and corresponding signed distances, one per object. During one training step of one mesh, we randomly sample 1,024 surface points for generating shape features through the encoder, and sample 16,000 query points for predicting signed distance values. We did not run a grid search for determining the optimal number of points. We experimented with a larger number for generating shape features (e.g., 5,000 instead of 1,024), which led to a decrease in generalization capabilities.

**Unlabeled Data**    For unlabeled data, we largely follow the preprocessing from [5] for generating points on and near the surface. For each input point cloud, we randomly sample 5,000 points and obtain $P = \{p_i \in \mathbb{R}^3\}_{i=1}^{5000}$. Then for each point $p_i$, we sample 20 Gaussian distributions each with mean 0 and standard deviation $\sigma$, where $\sigma$ is the distance between $p_i$ and its 50-th nearest neighbor. We add each distribution to $p_i$ to obtain 20 query points near the surface. Next, to ensure the model does not overfit to generated queries that only take up a small spatial proportion of the cube, we sample 30,000 additional points within the cube with each dimension ranging from -1 to 1. We then store the nearest neighbor $p$ to each query point $x$ by finding the shortest distance between $x$ and $p \in P$. In total, we have 5,000 points on the surface and 130,000 query points near the surface. During our first meta-learning stage, we randomly sample 1,024 surface points and 16,000 query points each training step, same as labeled data. During the second stage of semi-supervised training, we randomly sample 5,000 surface points and 5,000 query points each training step. More importantly, in this stage we train all 130,000 query points for each unlabeled object before moving on to the next one; that is, we train each unlabeled object in 130,000 / 5,000 = 26 steps. In [5], 20,000 points per input point cloud are sampled and 25 query points per surface point, but we reduce this number by a fraction to speed up training due to the higher number of training meshes.

## 1.2    Training Details

**Model Hyperparameters**    For Eqs. (3) and (9) (main paper) we the set the loss coefficients to $\lambda_m = \lambda_s = 0.1$, and $\lambda_p = 0.01$ in Eq. (7). $\lambda_m$ and $\lambda_s$ determine the weights of the unlabeled data and we found that $0.1$ was a good tradeoff between learning from unlabeled data and still mainly guiding the model with supervised, labeled data. A higher value such as $1.0$ would lead to accumulation of inaccuracies in the proxies of unlabeled data. The parameter $\lambda_p$ determines the weight of points on the surface vs points near the surface for unlabeled data. Here, setting $\lambda_p = 0.01$ helped with thinner structures, but a larger value led to gaps and holes in the reconstructions.

For our meta-learning stage, we define one epoch as sampling one object from one class. Since our labeled dataset consists of 20 classes, 20 objects are seen every epoch. We re-split the dataset into 10 labeled sets and 10 unlabeled sets every 1000 epochs and train for 40,000 epochs. We did not run search to determine the optimal value of split frequency or split ratio.

**Training Hyperparameters**    We use PyTorch [7] and train all models with the ADAM [8] optimizer. We set learning rates to $1 \times 10^{-4}$ with no decay.

For our meta-learning stage, we use 2995 meshes from 20 classes and the max batch size is 10 (each step trains one labeled and one unlabeled sample), so we set the batch size to 10. This takes up roughly 4.5 GB of GPU memory and we trained to 40,000 epochs in 2 days on an NVIDIA A100 GPU.

For our semi-supervised stage, one epoch is defined as sampling all objects once; i.e., each epoch iterates through 2995 labeled and 3408 unlabeled objects (see main paper for our specific data split).

Table 1: Decoder architecture. Following DeepSDF [4] with a number of changes, see text.

| Layer | In-Features | Out-Features | Notes |
|-------|-------------|--------------|-------|
| Linear | 259 | 512 | input = shape code (256) + xyz (3) |
| ReLU | | | |
| Linear | 512 | 512 | |
| ReLU | | | |
| Linear | 512 | 512 | |
| ReLU | | | |
| Linear | 512 | 512 | |
| ReLU | | | |
| Linear | 771 | 512 | skip connection = 512 + 259 |
| ReLU | | | |
| Linear | 512 | 512 | |
| ReLU | | | |
| Linear | 512 | 512 | |
| ReLU | | | |
| Linear | 512 | 512 | |
| ReLU | | | |
| Linear | 512 | 1 | output is not regressed with any activation |

Table 2: Encoder architecture. Following ConvOccNet [6]. "+" represents a stacking of operations. "lin" represents a fully-connected layer. "conv(a,b,c)" represents a 2D convolutional layer with kernel size a, stride b, padding c, and convT is a transposed convolution. "pool(a,b,c)" represents a 2D max pool with kernel size a, stride b, padding c.

| Layer Name | Type | In-/Out-Features |
|------------|------|------------------|
| Linear | lin | 3/128 |
| ResnetBlock1 | lin+ReLU+lin+ReLU+shortcut | 128/128 |
| ResnetBlock2 | lin+ReLU+lin+ReLU+shortcut | 128/128 |
| ResnetBlock3 | lin+ReLU+lin+ReLU+shortcut | 128/128 |
| ResnetBlock4 | lin+ReLU+lin+ReLU+shortcut | 128/128 |
| ResnetBlock5 | lin+ReLU+lin+ReLU+shortcut | 128/64 |
| Linear | lin+ReLU | 64/256 |
| DownConv1 | conv(3,1,1)+conv(3,1,1)+pool(2,2,0) | 256/32 |
| DownConv2 | conv(3,1,1)+conv(3,1,1)+pool(2,2,0) | 32/64 |
| DownConv3 | conv(3,1,1)+conv(3,1,1)+pool(2,2,0) | 64/128 |
| DownConv4 | conv(3,1,1)+conv(3,1,1) | 128/256 |
| UpConv1 | convT(2,2,0)+conv(3,1,1)+conv(3,1,1) | 256/128 |
| UpConv2 | convT(2,2,0)+conv(3,1,1)+conv(3,1,1) | 128/64 |
| UpConv3 | convT(2,2,0)+conv(3,1,1)+conv(3,1,1) | 64/32 |
| Final Conv | conv(1,1,0) | 32/256 |

Note that as mentioned previously, we train on all 130,000 query points for each unlabeled object. We train for 500 epochs. We set the batch size to 128 and train on an NVIDIA A100 for 5 days.

**Model Architecture** As described in the main paper, we use the encoder from [6] and the decoder from [4]. For the encoder, we set the latent size to 256 and hidden dimensions to 64. We use the modified PointNet [9] proposed in [6] that uses plane projection to learn local geometric features, and parallel UNets [10] to aggregate spatial information. For the decoder, we set the number of layers to 8, and use 512 hidden dimensions. We use ReLU activation after each layer except the last, and add a skip connection to the fourth layer. Different from [4], we do not use any normalization and we do not regress the final output with a Tanh activation. We also apply the geometric initialization

proposed in [11]. We provide a detailed list in Tab. 1 and 2. We also evaluate different architectures in Sec. 3.

## 1.3 Test-time Refinement

Most existing works that fit to multiple objects perform test-time refinement [4, 12, 11]. DeepSDF [4] creates a new latent code in 800 iterations while freezing the decoder. MetaSDF [12] performs 5 gradient steps. Notably, both methods use *ground truth signed distance* values for refinement. MetaSDF [12] showed some experiments of fitting only on point clouds, but reconstructions lack detail especially for intricate shapes.

The proposed self-supervised method allows us to perform test-time refinement on only the point cloud and no additional input. We sample 5,000 points per input point cloud. Sampling more points here, such as 15,000, leads to more detailed reconstruction outputs though the difference is not significant. Sampling size can be adjusted based on availability of data.

Using the 5,000 points, we then sample 20 queries per point as well as an additional 30,000 queries from the cube using the same method as in Sec. 1.1. Again, we emphasize that all these points are generated from the input point cloud and the process does not involve additional ground truths or annotations. Similar to DeepSDF [4], we perform refinement for 800 iterations. Each iteration, we randomly sample 5,000 query points. We set the learning rate to $1 \times 10^{-4}$. Finally, as we discuss in more detail in Sec. 5, we set the reconstruction level set to 0.005 due to inaccuracies in the unlabeled proxies (we set to 0.0 without refinement).

Our model is *also capable of reconstructing any target point cloud directly, that is without refinement,* and in the main paper we report numbers without refinement. We sample 1,024 points per point cloud, the same number we use during training. We show visualizations of reconstructions without refinement in Fig. 4. Our method still reconstructs details well although refinement is optimal for intricate shapes.

## 2 Additional Results

In the main paper we report numerical results of testing on 166 unseen categories. In supplemental Fig. 1, 2, and 3 we provide additional visualization results. Our testing method for both seen and unseen categories is the same as in Fig. 4 and 5 from the main paper but here we show a more diverse set of classes and shapes, including thin and hollow structures.

## 3 Alternative Network Architectures

We experiment with other network architectures and show visualizations in Fig. 4. First, we use the SIREN [13] architecture for our decoder. Since our point clouds only contain 3D coordinates and no normals, we exclude the loss term that requires normals. We also experiment with encoding coordinates with Fourier Features [14]. These experiments validate that our learning method is indeed applicable to different architectures, and, at the same time, validates the proposed architecture choice in our specific method.

## 4 Single-Object Methods

We next discuss single object methods which have been recently popular. We note that single object methods [15, 16, 17] solve a different problem than the proposed method. These methods focus on reconstructing single objects and scenes with detail, rather than generalization to multiple objects.

Nevertheless, we include an example of Ma et al. [17] and show results in Fig. 5. This method (Predictive Context Priors) reconstructs a *single unlabeled point cloud* with fine detail, including the gaps between the launchers at the side of the helicopter, but comes at the expense of learning shape priors and generalization capabilities. For completeness, we show a reconstruction of training three point clouds at once, though we note that this is not an intended setting for single object methods such as this method.

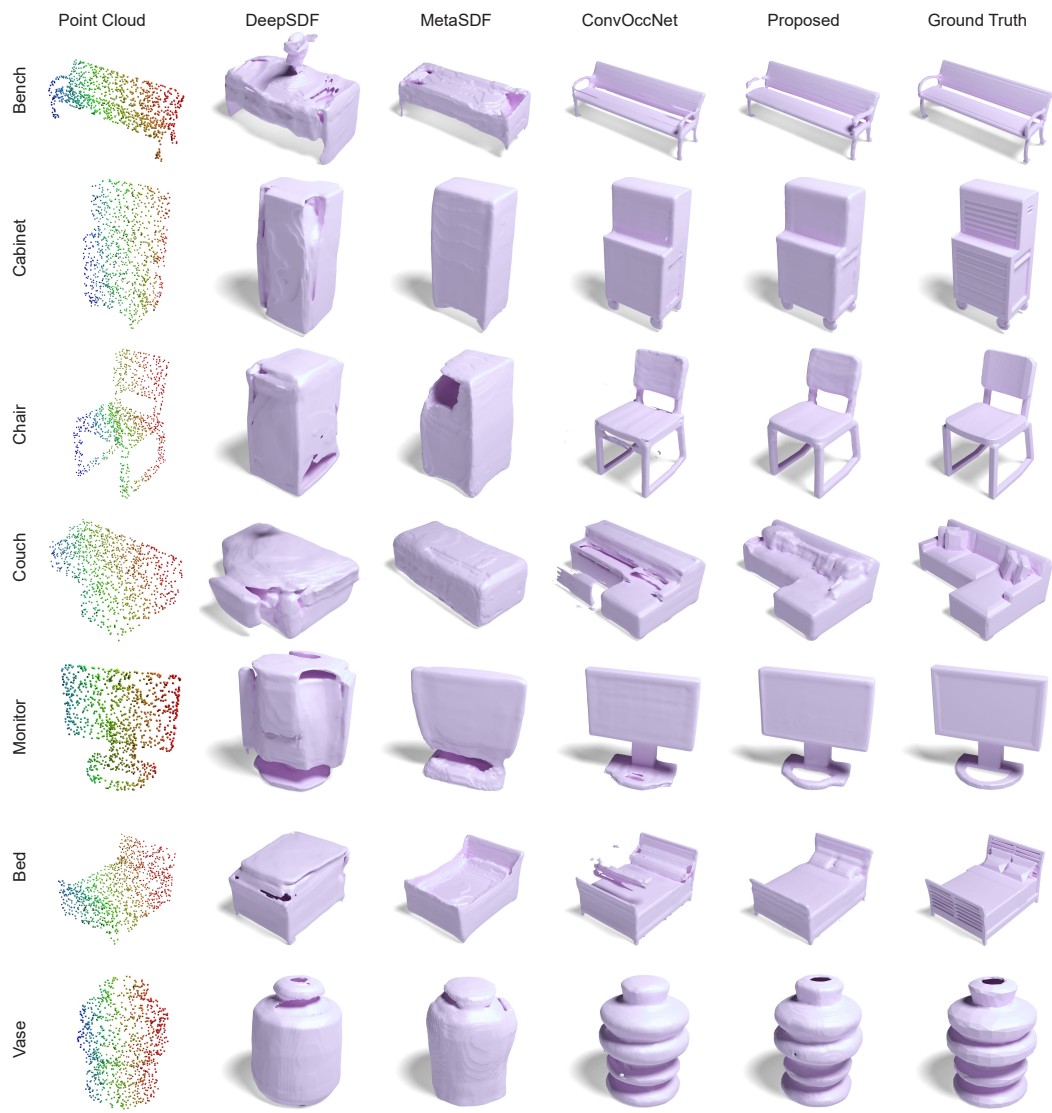

Figure 1: Additional reconstruction results on *seen* objects.

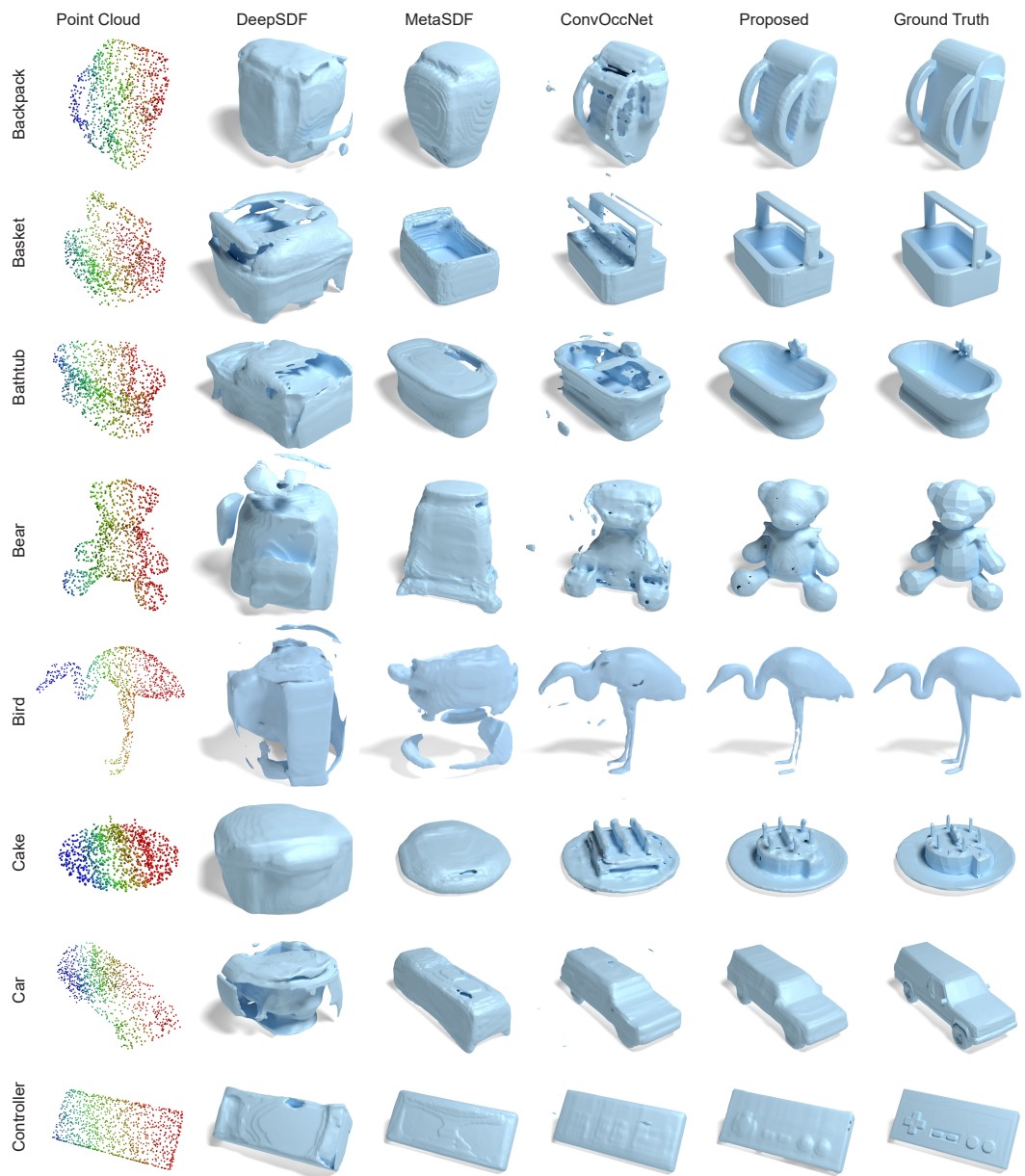

Figure 2: Additional reconstruction results on objects from *unseen* categories.

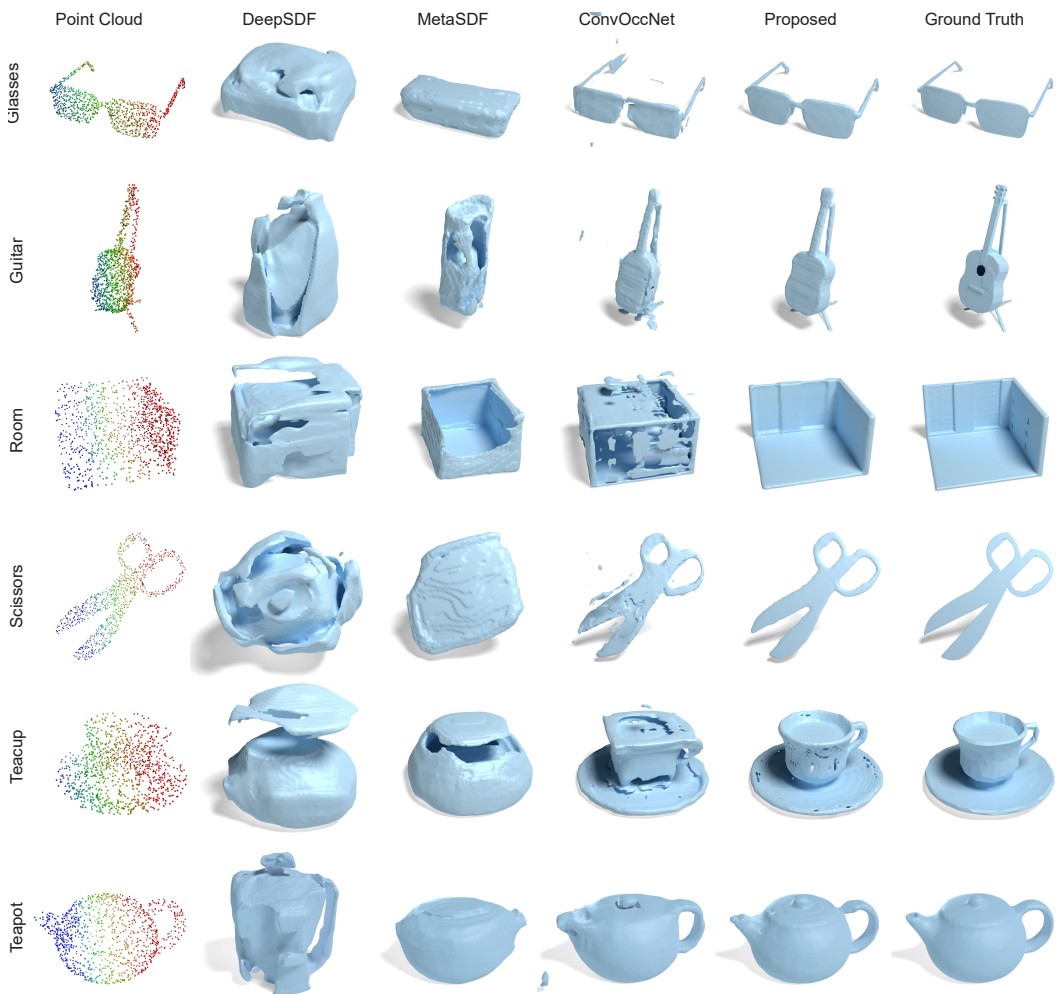

Figure 3: Even more reconstruction results on objects from *unseen* categories.

However, even for a single point cloud, Predictive Context Priors takes 30,000 input points and more than 10,000 iterations to train. On the other hand, our method reconstructs with high quality after just 800 iterations and 5,000 input points.

The main focus of Predictive Context Priors [17] and many other recent papers [15, 16] is different but complementary to ours. We leave the exploration of jointly achieving fine details with generalization across shapes as potentially exciting future work.

## 5 Unsupervised Methods for Training Unlabeled Data

Next, we analyze reconstructions of unlabeled data using our self-supervised method in comparison to existing unsupervised baselines: NeuralPull [5] and SAL [11]. The top row of Fig. 6 shows results of training and reconstructing objects from a single QueenBed class with 5 meshes. As a result of the explicit sign penalty, our method is able to reconstruct sharper details from irregular surfaces such as pillows. On the other hand, existing methods produce low losses for close approximations to the absolute values of the signed distances, rather than accurate sign predictions. Therefore, they tend to reconstruct a smoother curvature on shapes.

Using this experiment, we dive into a deeper comparison between our self-supervised loss and the approach described in NeuralPull [5]. First, we report accuracies of sign predictions between NeuralPull and our self-supervised method. We run inference on 130,000 points near the surface of

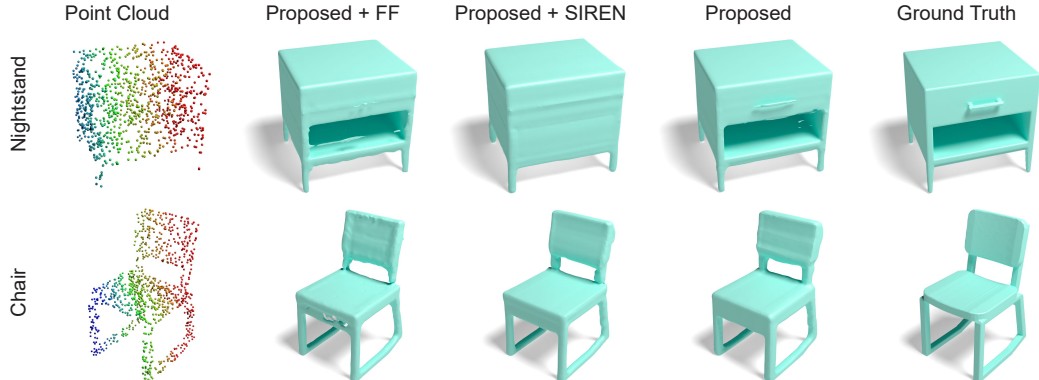

Figure 4: Testing on alternative architectures, including encoding coordinates using Fourier Features [14] and using the decoder of SIREN [13]. Results are *without* optimization or test-time refinement. These experiments validate that our learning method is applicable to different architectures.

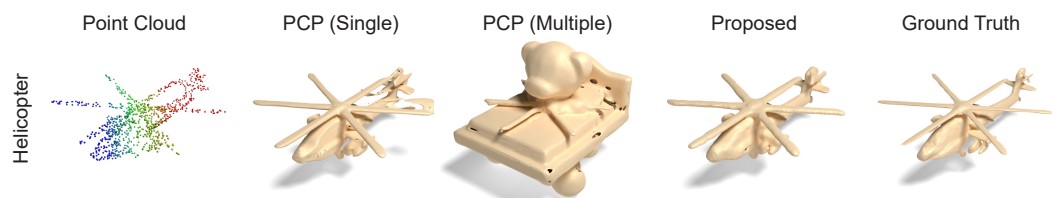

Figure 5: Comparing results to Predictive Context Priors (PCP) from Ma et al. [17]. PCP is designed to fit to a single object or scene with detail, taking 30,000 input points and 10,000 iterations. Our proposed method performs refinement using 5,000 input points and 800 iterations. PCP is not intended for training on multiple objects; we include PCP (multiple) which is trained on 3 objects only for completeness.

each of the five QueenBed point clouds (130,000 * 5 = 650,000 total). While both methods produced low $\ell_2$ losses, in Tab. 3 our method achieves higher accuracy in sign prediction.

Table 3: Confusion matrices for NeuralPull and our self-supervised loss sign predictions on five point clouds of the QueenBed class. Of the five point clouds, there are a total of 650,000 points to predict, with 449,650 being positive and 200,350 being negative.

| NeuralPull | Pred Positive | Pred Negative |
|---|---|---|
| 449,650 Pos | 37,779 (0.974) | 11,871 (0.026) |
| 200,350 Neg | 37,986 (0.190) | 162,363 (0.810) |

| Our SSL | Pred Positive | Pred Negative |
|---|---|---|
| 449,650 Pos | 449,245 (0.999) | 405 (0.001) |
| 200,350 Neg | 2,765 (0.014) | 197,585 (0.986) |

Despite good reconstruction results with small amounts of data, no purely self-supervised method works well when training on multiple classes. We randomly select 4 classes and 352 meshes and find that both NeuralPull [5] and the purely self-supervised variant of our method degrade quickly as the number of meshes increases. Interestingly, in this setting we found that the errors of using nearest neighbors rather than ground truth signed distance values accumulate and distort the original surface boundary. For fair comparison, we follow NeuralPull and use 0.005 level-set (rather than zero level-set) to represent the surface of the object. We assume the deviation in the positive direction a result of more training samples that are outside the surface since most surfaces we train on are convex.

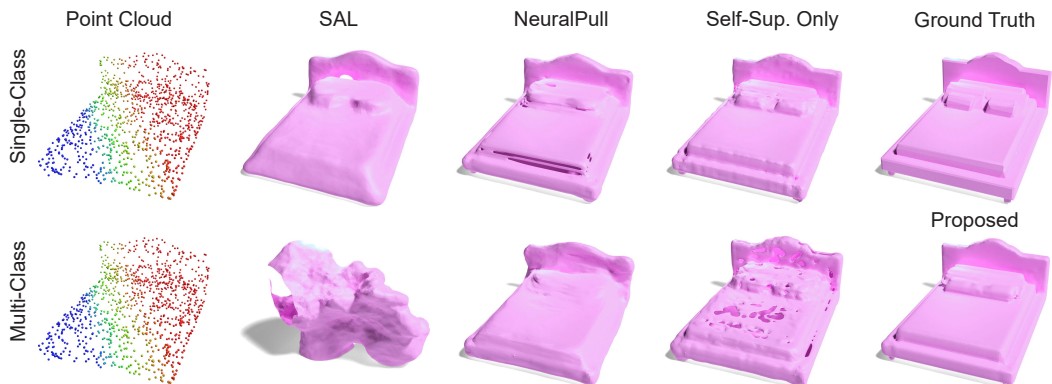

Figure 6: Comparing self/unsupervised methods. Our self-supervised loss penalizes incorrect signs and leads to sharper details from irregular surfaces such as pillows. Neither method works well when multiple categories are introduced. Our full proposed method includes labeled data for supervision which allows for scalability to multiple classes. All results are without test-time refinement.

This issue occurs also in Predictive Context Priors [17] which is unsupervised. Slight deviation from the zero level-set seems to be a problem with self/unsupervised methods; we leave this for future work. When trained with labeled data in our two-stage approach, we were able to reconstruct based on the zero level-set. However, after test-time refinement, the 0.005 level-set was required.

This supports our finding that methods without supervision are insufficient for tackling large-scale operations and are unable to take advantage of the availability of large amounts of unlabeled data. However, empirically we found that using our self-supervised method compared to existing unsupervised methods was more robust when trained along labeled data in our two-stage approach. When using NeuralPull [5] for unlabeled data, the model could not converge.

## 6  Effectiveness of Meta-Learning a Prior

In the main text, we mention there are a few advantages of meta-learning a prior. Compared to directly training on labeled and unlabeled data, our meta-learning process provides additional flexibility as we can tune the split frequency and ratio of labeled and "pseudo-unlabeled" data. The model repeatedly sees the same point clouds with and without labels, so shape features generated during labeled splits are shared with "pseudo-unlabeled" splits. In contrast, we find that semi-supervised training from scratch is unstable because errors in unlabeled data accumulate. Here, we provide two experiments to illustrate the effectiveness and limitations of our meta-learning approach.

**Training on diverse vs similar classes**   We train two models on our meta-learning first stage. The first trains on the following six diverse shape classes, with a mix of small, large, planar, convex, and concave surfaces: 'Bench', 'QueenBed', 'EndTable', 'FloorLamp', 'Monitor', 'PottedPlant.' The second trains on six classes that are semantically and geometrically similar, large planar shapes: 'EndTable', 'AccentTable', 'CoffeeTable', 'DiningTable', 'RoundTable', 'Table.' Then we evaluate on 166 unseen classes. The first model outperforms the second in reconstruction quality, validating that the diversity in the training set is more important than the raw number of classes. We do note that the difference in CD is noticeable but not very significant. This validates the effectiveness of our meta-learning approach: even with less diverse data, our model can still produce generalized priors. We provide quantitative results in Tab. 4.

Table 4: Results of training on six diverse and six similar classes. Mean/median Chamfer Distance (CD) of 166 unseen classes, scale is $10^{-4}$.

|  | Six Diverse Classes | Six Similar Classes |
| --- | --- | --- |
| CD | 0.972 / 0.657 | 1.321 / 0.934 |

**Training only with the supervised loss** We train two separate models for a prior. Both train on six classes: 'Bench', 'QueenBed', 'EndTable', 'FloorLamp', 'Monitor', 'PottedPlant,' with a total of 600 meshes. The first model is trained using our two-stage learning approach. The second model is trained with only the supervised loss, but introduces dropout (20%) and small amounts of Gaussian noise perturbations to the point clouds to reduce overfitting.

Then, we use both priors to train our second stage in exactly the same fashion. With our prior, our model performs significantly better when reconstructing unseen classes. We provide quantitative results on 166 unseen classes in Tab. 5. We highlight that after training stage 1, both methods perform on par on unseen classes. However, our prior that emulates semi-supervised training is able to significantly outperform in the second stage, where we introduce true unlabeled data samples.

Table 5: Comparison of our approach to a prior trained only with the supervised loss. Mean/median Chamfer Distance (CD) of 166 unseen classes, scale is $10^{-4}$.

|  | Proposed | Supervised Prior (Augmentation + Dropout) |
|---|---|---|
| After stage 1 | 0.972 / 0.657 | 1.027 / 0.697 |
| After stage 2 | 0.420 / 0.316 | 0.851 / 0.568 |

# 7 Reproductions and Licenses

All datasets we use and existing methods we compare to have released their code as open source. For fair comparison, we make no changes to released code except for the dataloader for loading our datasets. We test on the same hardware platform. We will also release our own code as open source under the MIT license.

# 8 Limitations

There are a few limitations of our work that we would like to address in future work. First, we need multiple categories to learn a diverse set of geometry. Training on just one category (e.g., chairs) will not allow the model to generalize to multiple intricate shapes. Each category should also contain a certain number of representative objects. We did not run experiments to determine the number of categories required, but in our work we choose 20 categories, each with at least 80 objects, which worked out well.

Second, for intricate details, test-time refinement is required. The downside of test-time refinement is that each object requires more time to reconstruct. Fortunately, our self-supervised method allows for refinement without additional inputs other than the point cloud.

Finally, our method trains and tests on full-view point clouds. In future work, we would experiment with single or partial-view point clouds for shape completion tasks.

# 9 Broader Impact

Our method has no direct ethical impact, but potentially could be used for 3D reconstruction tasks in face recognition and human reconstruction, which could result in privacy issues or gender and racial biases.