# OpenReview forum: "GenSDF: Two-Stage Learning of Generalizable Signed Distance Functions"
_NeurIPS.cc/2022/Conference — NeurIPS 2022 Accept_

### Official Review · Reviewer_39TN · 2022-07-07

**Rating:** 4
**Confidence:** 5
**Soundness:** 3 good
**Presentation:** 3 good
**Contribution:** 3 good

**Summary:**

The main contribution of this paper is to improve the generalization capabilities of implicit representations learning. This paper introduce an episodic training scheme where we split data into subsets with disjoint classes to learn a generalized shape representation at the first stage. The second stage allows the model to ingest large amounts of unlabeled data to diversify the shape model learned in the first stage.

**Questions:**

(1) In the first stage, every f epochs, this paper splits data X into two subsets with disjoint categories, and then train both data with self-supervised loss and supervised loss respectively. According to my understanding, training the network with ground-truth signed distance values supervision through the supervised loss will give more accurate shape priors. I do not understand the role played by the self-supervised loss here. Even though the paper gives a vague explanation: “episodic training on non-overlapping categories forces the model to develop generalized representations as it cannot overfit to the data.” If the self-supervised loss simply prevents overfitting to the data, there are many common training strategies in the field of machine learning  that can prevent overfitting. A more convincing proof of principle for the self-supervised loss is needed here.

(2) I am puzzled by the results of Table 1, whether such a large improvement can be achieved using only unlabeled point cloud data, and what is the reason for the improvement? How is the prediction of /phi(x) guaranteed to be correct if it does not depend on the labeled point cloud. The loss function of Eq. 7 can be equally reduced if it is predicted /phi(x) to all positive or negative signs in Eq. 8. Both SAL and NeuralPull give proofs that they can predict the correct sign, and this paper is unconvincing in this part.

**Ethics Review Area:**

["I don’t know"]

**Limitations:**

I don't foresee any potential negative societal impacts.

**Strengths And Weaknesses:**

Strengths：

(1) This paper proposes a training scheme that can learn implicit representations from both labeled and unlabeled point cloud data, and the method produces effective experimental results.

(2) Enhancing the implicit representation learning capability of the network using both labeled and unlabeled point cloud data simultaneously is a meaningful research problem, and from the experimental results, the paper also proposes a framework that can combine both kinds of data effectively.

Weaknesses：

Even though I am knowledgeable about works related to implicit representation learning, I have difficulty understanding the reasons why the framework proposed in the paper works outperforms existing neural SDF methods. No convincing explanation is given in the paper for this proposed framework to achieve such a large effect of enhancement.

---

> ### Author Response · Authors · 2022-08-02
> **Response to Reviewer 39TN**
>
> > Response to Question 1
>
> We employ the self-supervised loss in the first stage to emulate second stage training. In our second stage, we introduce unlabeled data which mandates unsupervised training objectives. We find that this emulation in the first stage is essential to consuming real unlabeled data in the second stage.
>
> To better illustrate this, we point to the following experiment:
> We train two different models for a prior on six classes with a total of 600 meshes: the first model is trained with the proposed method while second model is only trained with the supervised loss. As suggested by the reviewer, for the second model we introduce existing strategies against overfitting. We use dropout (20%) and add small amounts of Gaussian noise perturbations to the point clouds.
>
> Then, we use both priors to train our second stage in exactly the same fashion. Our method performs better when reconstructing unseen classes. This validates that emulating the desired training process improves generalization. We provide quantitative results on *unseen* classes (mean/median CD * 10-4):
>
> |  | Ours   | Supervised-only Prior (Augmentation + Dropout) |
> | --------        | --------        | --------      |
> | After training stage 1| 0.972 / 0.657 | 1.027 / 0.697|
> |After training stage 2| 0.420 / 0.316  | 0.851 / 0.568|
>
> We highlight that after training stage 1, both methods perform approximately on par on unseen classes but not after stage 2, validating the effectiveness of the two stage approach.
>
> > Response to Question 2 (and weaknesses)
>
> Theorem 1 from NeuralPull provides a proof that, **if** their choice of loss converges, then it will converge to a *signed* distance function. It provides no guarantees on convergence, nor mathematical intuition for training behaviour given varying object geometries, finite sampled point clouds, and MLP initialization. Indeed *both our approach and NeuralPull are reliant on SAL’s geometric initialization* to avoid noisy non-converging solutions.
>
> Analytically, the difference between our formulation and NeuralPull's is that they approximate the normal of points on the latent surface $\textbf{p}$ as $\textbf{N} = \nabla f(\textbf{q})/|\nabla f(\textbf{q})|_2$, where $\textbf{q}$ is a query point, proportional to the gradient of the MLP-learned SDF. Note that this only becomes true for the unknown target SDF. In contrast, we estimate the normal of $\textbf{p}$ as a signed unit vector to a query point $\textbf{q}$. Specifically, both methods predict points $p^\prime_i$ on the surface as:
>
> NeuralPull: $p^\prime_i = q_i - f(q_i) \times \frac{\nabla f(q_i)}{||\nabla f(q_i)||_2}$
>
> Ours: $p^\prime_i = q_i - |f(q_i)| \times \frac{(q_i - p_i)}{||(q_i - p_i)||_2}$
>
> Both methods approximate the normal of $\textbf{p}$. However, during training, the surface is represented by a finite set of random sampled locations. Thus, for a given query point $q$, NeuralPull is not guaranteed to find a surface point $p$ such that $\nabla f(q)$ is the direction vector between $p$ and $q$, and is susceptible to accumulated errors from this procedure. As such, our formulation is an alternative of NeuralPull's formulation that accounts for non-continuous sampling, improving convergence behavior without disputing the proof of Theorem 1.
>
> Eq. (8) of our main text provides an intuitive explanation of why our method penalizes incorrect sign predictions. Empirically, we further illustrate the effectiveness of our approach with the following two experiments below:
>
> First, we run NeuralPull and our self-supervised loss as stand-alone methods. We train on five point clouds from the QueenBed class, and run inference on 130,000 points near the surface of each point cloud. While both methods produced low $\ell_2$ losses, our method produced significantly higher sign prediction accuracy, which explains why our reconstruction results have sharper edges and details, while NeuralPull's appear often smoothed out (Fig. 8 in Appendix of revised manuscript).
>
> Confusion matrix for NeuralPull:
> | 650,000 points  | Pred Pos   | Pred Neg |
> | --------        | --------        | --------      |
> | 449,650 Pos| 437,779 (0.974) | 11,871 (0.026)|
> | 200,350 Neg| 37,986 (0.190)  | 162,363 (0.810)|
>
> Our self-supervised method:
> | 650,000 points  | Pred Pos   | Pred Neg |
> | --------        | --------        | --------      |
> | 449,650 Pos| 449,245 (0.999) | 405 (0.001)|
> | 200,350 Neg|  2,765(0.014)   | 197,585 (0.986)|
>
> Second, we swap out our self-supervised loss for NeuralPull's loss to train our full model. We provide training details, curves, and results in the Appendix (Fig. 9, Tab.4) of our revised submission.
>
> Lastly, SAL provides rigorous theoretical foundations on the amount of information in unlabeled point clouds, namely that you can *provably* fit an SDF to these point clouds alone. While these proofs cannot be directly extended to our mixed-loss meta-learning setup, we will add further reference and discussion of them in the final text.

---

> > ### Author Response · Authors · 2022-08-05
> > **Follow-up on Rebuttal**
> >
> > Dear Reviewer 39TN,
> >
> > Following your questions, we addressed the role of the self-supervised loss in our meta-learning approach, convergence proofs, and we provided additional intuition and experiments to illustrate the effectiveness of our approach. To this end, we experimentally validate the sign prediction accuracy of our self-supervised loss. In light of this, we would like to know whether you believe we have addressed your concerns, and if so we hope that you would be willing to increase your score.
> >
> > Thank you for your time,
> >
> > The Authors

---

> > ### Comment · Reviewer_39TN · 2022-08-09
> > **Foundation theory**
> >
> > Thanks for the authors of the reponse to my questions. I endorse the experimental results provided by the authors, which demonstrate the effectiveness of this method. For me, it would make the paper better if the underlying theoretical proofs were provided for the self-supervised loss and convergence to the signed distance function.

---

### Official Review · Reviewer_GFH2 · 2022-07-11

**Rating:** 6
**Confidence:** 4
**Soundness:** 3 good
**Presentation:** 3 good
**Contribution:** 4 excellent

**Summary:**

Given multiple-category labeled (with GT SDF) and unlabeled point cloud data,  this paper introduces a simple yet effective meta-learning inspired pretraining stage before the conventional semi-supervised stage in order to train a generalizable SDF function model.  Their strategy frequently randomly splits the labeled data into a self-supervsied and GT-supervised group with different categories, which creates diversity of different tasks and encourage a shared embedding network for better model initialization, and better leverage labeled multi-category data as guidance, assisted by self-supervision loss on unlabeled data.



**Questions:**

1. What if in stage 1, we simply first train on all the labeled data directly with GT/SSL loss, what will be the results on the next stage of semi-supervised training? The no split in Table 2 seem to be the desired one, but I am not sure whether it is followed by semi-supervised training, and have the exactly same training setting except for the splitting;

2. How to decide the split ratio between labeled and unlabeled dataset, would smaller number of labeled data (like only 5 classes instead of 20/1) lead to big degrade to the methods? How is the split frequency being decided, will it be a big influence factor to the pretraining performance?

3. A recent paper 'Shape As Points: A Differentiable Poisson Solver' can also reconstruct surface from complete point cloud by learning a general geometry equation solver, is it possible to also compare with this paper? Although quantitative results would be better, comparison in text is valid if the experiments become too complex.

**Limitations:**

The main limitations are two-fold as mentioned by the authors:
1. It requires multi-category data with ground truth for training;
2. It does not consider partial scans, and may fail on partial and noisy data;

**Strengths And Weaknesses:**

**Strengths**\
[originality]
It is novel to apply episodic scheme on randomly split hybrid training for SDF learning as pretraining;
They further introduce a self-supervised loss regularization item[Section 3.2], which is well considered for the target task;

[quality & clarity]
The paper is written with a smooth logic flow, and clear explanation on the methods and experiments, with a lot of high-quality visualization and comparison, like Figure 5 in main paper, and figures in the supps.

[significance]
This simple yet effective training paradigm is helpful and inspiring for a lot of 3D learning tasks, the achieved results[Figure 3, Table 1] are
much better than compared baselines.

**Weakness**
1. This paper gives me an illusion of as long as you can keep randomly splitting your labeled multi-category dataset
and train with two different losses(GT-supervised and Self-supervised), you can get a well-generalized model with good pretraining weights;
I am surprised at this conclusion, in which the author might want to give more details like training curves, failure cases, to really show the full potential and limitation of this proposed framework, while also explain better on the intuition behind;

2.  No ablation study on the discussed 'Self-supervised Loss Component' in line 163-182, we want experiments to support the authors' argument on this alternative formulation to avoid incorrect sign predictions;

3. As mentioned by the author in the supp., it seems current work only supports complete point cloud, it doesn't generalize to real-world partial scans, which is the major and crucial application scenerios for semi-supervised learning/Meta-learning to demonstrate its effectiveness. Even though current work is not designed for partial scans, the author should at least show some qualitative results.

---

> ### Author Response · Authors · 2022-08-02
> **Response to Reviewer GFH2**
>
> > “This paper gives me an illusion of as long as you can keep randomly splitting your labeled multi-category dataset and train with two different losses(GT-supervised and Self-supervised), you can get a well-generalized model with good pretraining weights”
> >
> The specific dataset split and unsupervised loss is essential for the proposed approach to work. We illustrate this with the following experiments that provide further intuition. In the first experiment, we show that the diversity in data is important for the model to learn a generalized prior. Randomly splitting is an effective strategy if the given dataset is balanced in terms of geometry. In the second experiment, we show that not any two (supervised/unsupervised) losses work but the choice of the unsupervised loss is essential for the convergence of the proposed method.
>
> #### Training Data in First Stage:
> We train two models on our meta-learning first stage. The first trains on the following six diverse shape classes: 'Bench', 'QueenBed', 'EndTable', 'FloorLamp', 'Monitor', 'PottedPlant'. The second trains on six classes that are semantically and geometrically similar: ‘EndTable’, 'AccentTable', 'CoffeeTable', 'DiningTable', 'RoundTable', 'Table'. Then we evaluate on 166 unseen classes. The first model outperforms the second, validating that the diversity in the training set is more important than the raw number of classes. Although even with less diverse data, our meta-learning approach can still produce generalized priors.
>
> We provide quantitative results (mean/median CD * 10-4) here and training curves in *Fig. 11 of our revised submission*.
>
> |   | Six Diverse Classes   | Six Similar Classes |
> | --------        | --------        | --------      |
> | CD| 0.972 / 0.657 | 1.321 / 0.934|
>
> #### Effect of Unsupervised Loss:
> To illustrate the importance of our unsupervised loss, we point to Sec. A.2 in our revised paper and Fig. 6 of our supplemental material. Using NeuralPull's loss instead of our unsupervised loss, training seems to converge to local minima. We attribute this to inaccurate sign predictions, which build up as we increase the number of meshes. This conflicts with the supervised loss which produces accurate sign predictions. Thus, the model is unable to converge at a meaningful signed distance function. We provide training curves in *Fig. 9 of our revised submission* and a qualitative example in *Fig. 10*. Quantitative evaluation (mean/median CD * 10-4) is listed below:
>
> |   | Ours (stage 1)   | Ours w/ NeuralPull loss (stage 1) |
> | --------        | --------        | --------      |
> | CD| 0.972 / 0.657 | 359.6 / 361.0|
>
> > “current work...doesn't generalize to real-world partial scans, which is the major and crucial application scenerios for semi-supervised learning/Meta-learning”
>
> While we demonstrate robust reconstruction of real-world complete point cloud scans from YCB (Fig. 6 and 7), we are not able to directly apply our method to partial scans. Our self-supervised loss for training on unlabeled data relies on nearest neighbors on the point cloud as proxies for ground truth distance values, which missing regions in partial scans lack. We agree that this is an important direction of future work and will discuss this in the next revision of the text.
>
> > “What if in stage 1, we simply first train on all the labeled data directly with GT/SSL loss, what will be the results on the next stage of semi-supervised training?”
>
> *Directly with GT loss:* Our “no split” experiment is not followed by semi-supervised training. A prior that is trained only on the supervised loss performs worse on the second stage, because there is no emulation of the model to train on unlabeled data. We validate this further as follows:
>
> We train two separate models for a prior. Both train on six classes with a total of 600 meshes. The first model is trained exactly using our method as explained in our manuscript. The second model is trained only with a supervised loss, using dropout (20%) and small amounts of Gaussian noise augmentation to the point clouds to prevent overfitting.
>
> Then, we use both priors to train our second stage in exactly the same fashion. With our prior, our model performs significantly better when reconstructing unseen classes. We provide quantitative results on 166 *unseen* classes (average/median CD * 10-4) here:
>
> |  | Ours   | Supervised Prior (Augmentation + Dropout) |
> | --------        | --------        | --------      |
> | After training stage 1| 0.972 / 0.657 | 1.027 / 0.697|
> |After training stage 2| 0.420 / 0.316  | 0.851 / 0.568|
>
> We highlight that after training stage 1, both methods perform on par on unseen classes. However, our prior that emulates semi-supervised training is able to excel in the second stage.
>
> *Directly with SSL loss:* We report in supplemental Sec.5 that using SSL alone cannot produce a well-trained prior and cannot be used to train the second stage.

---

> > ### Author Response · Authors · 2022-08-07
> > **Follow-up on Rebuttal**
> >
> > Dear Reviewer GFH2 ,
> >
> > Following your questions, we provided additional experiments to illustrate both the potential and limitations of our meta-learning approach. We also empirically showed the sign prediction accuracy of our self-supervised formulation. In light of this, we would like to know whether you believe we have addressed your concerns, and if so we hope that you would be willing to increase your score.
> >
> > Thank you for your time,
> >
> > The Authors

---

> > > ### Comment · Reviewer_GFH2 · 2022-08-08
> > > **Reply to rebuttal**
> > >
> > > Hi,
> > >
> > > Thanks for providing the detailed ablation results on data split, different training setting, and etc.  Although the rebuttal has addressed most of my questions, my major concern is still on the significance of the major technical contribution, requirements of massive multi-category labeled training data, and its generalization to real-world data. To be more specific, the author might be able to give a little bit more clarification that how many labeled instances per category would be enough, like would 10/20 instances per category be still a reasonable setting? I do see that the generalization to real-world data is actually in a reasonable quality, but would the proposed framework can be combined with further test-time mesh refinement?
> > >
> > > Thanks!

---

> > > > ### Author Response · Authors · 2022-08-09
> > > > **Response to Reviewer GFH2 (2/2)**
> > > >
> > > > A few dozen point clouds is insufficient for our model to learn a strong generalized prior. The remaining time in the discussion phase did not allow us to report results on 10 or 20 instances per category. In the final version, we will analyze both training stages for varying instance counts per category.
> > > >
> > > > We highlight that our method generalizes to unseen, real-world distributions given *only Acronym, a synthetic dataset*. There are a number of existing datasets that include even larger instance counts than Acronym. ShapeNetSem (a superset of Acronym) has 270 categories and 12,000 mesh instances. The ABC dataset has over 1 million meshes. ModelNet includes 151,128 mesh instances. Our method produces state-of-the-art results on unseen classes using 2,995 labeled meshes and 3,408 unlabeled point clouds. Specifically, using only this synthetic training data, we demonstrate results on the YCB dataset (see Fig. 7 of the revised submission) which is a real-world point cloud dataset acquired with RGBD cameras. Our mesh reconstructions of the measured data, e.g., recovering the "handle" of a pitcher, may serve as input to complex robotic grasping tasks. Importantly, existing methods fail for this task (see Fig. 7 of the revised submission). Even on synthetic, unseen data, existing methods fail (see Fig. 5).
> > > >
> > > > Addressing the second question, we can incorporate test-time refinement by using our self-supervised loss to refine the model to a given input point cloud. We provide more specific details in Sec. 1.3 of our supplement.

---

### Official Review · Reviewer_84Fy · 2022-07-11

**Rating:** 7
**Confidence:** 4
**Soundness:** 2 fair
**Presentation:** 4 excellent
**Contribution:** 4 excellent

**Summary:**

The paper considers the task of learning to reconstruct shapes (via the SDF) from "unlabelled" point clouds (no mesh and no normals) using a single network that must generalize for unseen shapes (and shape classes). The network is allowed to train on both labelled data (meshes) and unlabelled data (unoriented point clouds). The proposed method trains with a standard supervised loss on labelled data, and a self supervised loss that builds upon Neural Pull [19] (but corrects it to properly penalization incorrect signs) for unlabelled data. They further train the supervised loss with the self-supervised loss on psudo-unlabelled data, claiming it greatly helps generalize, with ablations studies demonstrating this.

**Questions:**

Why does adding the semi-supervised training decrease the Chamfer distance on seen classes so much (0.503 to 0.351)? Shouldn't the network have been trained enough to memorise/overfit those shapes/shape classes? What do you believe the semi-supervised training is adding here (we don't need object level generalisation to perform well on seen shapes)?

**Limitations:**

A short discussion of limitations is given in the conclusion. No negative societal impacts are discussed, but I agree that there are no direct negative impacts.

**Strengths And Weaknesses:**

Strengths
- A data driven approach to shape reconstruction with a strong focus on generalization
- Specifically allows to learn to generalize from a large corpus of unlabelled shapes.
- Fixes an important weakness in the NeuralPull training objective
- Proposes introducing unsupervised training into the the initial supervised training step that shows significant improvement in generalization
- Shows large improvement over baseline methods for both unsupervised and supervised methods, without further test time refinement

Weaknesses
- The baselines for this task could be better.
	- Section 4.4 says that "No split" is essentially ConvOccNet but has lower CD due to a different choice of decoder, but the difference from this choice as decoder is really large. As a result one of the baselines for Section 4.1 should be ConvOccNet with the same decoder as you use.
	- The method is technically given access to 76 more classes via unsupervised training for it to learn to generalize on (since the other methods do not have an unsupervised learning method), while supervised methods only get 20. It would be better to try supervised training on more diverse classes (but a few in each class such that the overall number of supervised meshes to train on is the same) to better emulate a supervised training approach that is trying to generalize to other shape classes.
	- It would be interesting to have a baseline that does the same setup as your approach (Phase 1 and 2) but change the self-supervised loss to be the same as Neural Pull, which would show how important your change is.
	- For unsupervised methods (Section 4.2), it would be good to compare to IGR without normals (the autodecoder setting), which has a better unsupervised loss (Eikonal term) than SAL and doesn't have the defect that NeuralPull does. For an even stronger baseline, PHASE+FF [* 1] or DiGS [* 2] would be even better (again with the autodecoder setting).
- No quantitative comparison to single network per shape methods (which you call Single-object methods in the supp material), e.g. IGR, SIREN. Although, as you argue in the supp material, this is a different setting, such quantitative results (which you have given a qualitative result for) would serve as a good upper bound for seeing how close your method can generalize.

[* 1] https://arxiv.org/abs/2106.07689
[* 2] https://arxiv.org/abs/2106.10811

---

> ### Author Response · Authors · 2022-08-02
> **Response to Reviewer 84Fy**
>
> > “one of the baselines for Section 4.1 should be ConvOccNet with the same decoder as you use."
>
> The "No split" setting is indeed exactly ConvOccNet with our decoder (encoder of ConvOccNet + our decoder), and we have clarified this in the revised manuscript. We validate the effectiveness of the proposed decoder quantitatively in Tab. 2 as an ablation experiment.
>
> > “The method is technically given access to 76 more classes...while supervised methods only get 20. It would be better to try supervised training on more diverse classes...to better emulate a supervised training approach that is trying to generalize to other shape classes.”
>
> We address this question in two parts.
>
> #### Comparison to supervision with more classes:
> We report three results below: training our meta-learning approach with 20 classes, and training only with the supervised loss on 20 and 76 classes. We test all methods on 166 unseen classes. With our meta-learning approach, our model achieves lower CD even when compared to training 76 classes using only the supervised loss.
>
> |   | Ours (Stage 1), 20 Classes   | Supervised Only, 20 Classes | Supervised Only, 76 Classes |
> | --------        | --------        | --------      |------|
> | CD| 0.407 / 0.250 | 1.164 / 1.367 | 0.908 / 0.866 |
>
> #### Training proposed method with more classes in the first stage:
> As 20 classes appear to be 'enough' for the model to learn a generic shape prior, to better understand *what* exactly in the training data leads to better generalization, we ran the following experiment:
>
> We train two models on our meta-learning first stage. The first trains on the following six diverse shape classes: 'Bench', 'QueenBed', 'EndTable', 'FloorLamp', 'Monitor', 'PottedPlant'. The second trains on six classes that are semantically and geometrically similar: ‘EndTable’, 'AccentTable', 'CoffeeTable', 'DiningTable', 'RoundTable', 'Table'. Then we evaluate on 166 unseen classes. The first model achieves a lower CD than the second, validating that the diversity in the training set is more important than the raw number of classes.
>
> |   | Six Diverse Classes   | Six Similar Classes |
> | --------        | --------        | --------      |
> | CD| 0.972 / 0.657 | 1.321 / 0.934|
>
> > “It would be interesting to have a baseline that does the same setup as your approach (Phase 1 and 2) but change the self-supervised loss to be the same as Neural Pull”
>
> We swap out our self-supervised loss for NeuralPull's loss to train our full model. We only train six classes on stage 1 due to time constraints. With NeuralPull's loss, training seems to converge to local minima. We attribute this to inaccurate sign predictions, which build up as we increase the number of meshes. We provide training curves in *Fig. 9 of our revised submission* and a sample of reconstruction results in Fig. 10. For a more detailed discussion of sign predictions, see *Tab. 3 of our revised submission*. Quantitative results (mean/median CD * 10-4) are reported below:
>
> |   | Ours (stage 1)   | Ours w/ NeuralPull loss (stage 1) |
> | --------        | --------        | --------      |
> | CD| 0.972 / 0.657 | 359.6 / 361.0|
>
> > "For unsupervised methods (Section 4.2), it would be good to compare to IGR...PHASE+FF [* 1] or DiGS [* 2]"
>
> We were unable to complete this experiment due to time constraints. In our final version, we will compare to these methods.
>
> > “No quantitative comparison to single network per shape methods...e.g. IGR, SIREN...such quantitative results...would serve as a good upper bound for seeing how close your method can generalize.”
>
> We have conducted an additional experiment to compare our method to a single-object method, SIREN, and will include additional results in the final version. These methods can outperform ours on single objects (lower CD) because they have more complex or specific training objectives and loss functions. However, we note, that single object methods fail for multiple objects, as shown in Fig. 5 and 6 in our supplemental material. We run SIREN on the author provided sample point cloud, "Thai Statue," and also use our model to run inference on the same file.
>
> |   | Ours   | SIREN |
> | --------        | --------        | --------      |
> | CD of Thai Statue| 0.124 | 0.064|
>
> > “Why does adding the semi-supervised training decrease the Chamfer distance on seen classes so much (0.503 to 0.351)? Shouldn't the network have been trained enough to memorise/overfit those shapes/shape classes?”
>
> Our goal is generalization, so we employed early stopping to prevent our first stage from overfitting on the labeled set. After training for double the number of iterations, our first stage is able to achieve improved results on seen classes (CD of 0.351), but performs worse in generalization. This means for the "seen shape" evaluation, semi-supervised training plays a role of extending training.

---

> > ### Author Response · Authors · 2022-08-07
> > **Follow-up on Rebuttal**
> >
> > Dear Reviewer 84Fy ,
> >
> > Following your questions, we provided additional baseline experiments to illustrate the effectiveness of our approach. In light of this, we would like to know whether you believe we have addressed your concerns, and if so we hope that you would be willing to increase your score.
> >
> > Thank you for your time,
> >
> > The Authors

---

> > > ### Comment · Reviewer_84Fy · 2022-08-08
> > > **Response to Follow-up**
> > >
> > > Thank you for your detailed response.
> > >
> > > My questions have been sufficiently answered by the authors, and the new results confirm the claims made in the paper. I think that having first set of results (_Ours (Stage 1), 20 Classes_; _Supervised Only, 20 Classes_; _Supervised Only, 76 Classes_) in the main paper would be a useful addition, the rest can go to supplementary.
> > >
> > > I have also read the other reviews and the authors' reponses to them.
> > >
> > > I stand with my original rating and strongly recommend accepting the paper.

---

### Official Review · Reviewer_8obm · 2022-07-11

**Rating:** 6
**Confidence:** 4
**Soundness:** 3 good
**Presentation:** 3 good
**Contribution:** 3 good

**Summary:**

This work proposes GenSDF, a method for object surface estimation, parameterized by SDF, from 3D point cloud inputs. The main distinctions of GenSDF to prior work is its ability for high quality generalization to novel classes not seen during training, and the ability to take advantage of "unlabeled" point cloud data (without know SDFs) as supervision during training. GenSDF achieves this by two well designed training phases, first a meta learning procedure to train a general shape prior, by sampling labelled and unlabeled subsets of the labelled training set, and a second semi-supervised step where the fully labelled training set is used in addition to an additional entirely unlabeled set. GenSDF also proposes a well designed loss for using unlabeled point clouds as supervision for an SDF prediction model.

Evaluation is performed on Acronym (a subset of ShapeNet), split into 20 labeled training classes, 70 unlabeled classes for semi-supervised training, and 166 classes for testing. GenSDF significantly outperforms SDF-supervised and unsupervised . GenSDF can also directly generalize to point clouds from scans of the YCB dataset. Ablation studies directly demonstrate the benefit of the proposed two-stage training procedure.

**Questions:**

1. It is clear that GenSDF is better at generalization to unseen categories than prior methods. However, the shape similarity of what are supposed to be different categories in ShapeNetSem affects these claims to some extent. Can the authors please comment on how the fact that there are semantically very similar categories that may end up being split in the seen and unseen sets affects the claims that can be made about generalization ability?
2. Regarding the point under weaknesses in significance, can the authors please provide a bit more context about how they design the evaluation task relative to potential real world applications of GenSDF? Including this in the introduction of the paper would help the reader understand the significance of what the model can do.

**Limitations:**

The authors have discussed some limitations of their work throughout the draft. The paper itself doesn't have high potential for negative societal impact.

**Strengths And Weaknesses:**

## Originality
### Strengths
Direct generalization to unseen categories in the domain of surface estimation from point clouds has not been investigated before, and is an important application (see Significance). This method proposes a novel and well designed two stage training procedure, along with a loss function for using unlabeled data samples. The model's design is supported by high quality ablation studies and significantly higher performance than baselines in terms of chamfer distance.

## Quality
### Strengths
1. _Simple, elegant and effective implementation_: The high level idea of learning a general shape prior that can be used for semi supervised training with unlabeled data makes sense, and paper addresses this using a simple but effective approach of repeatedly partitioning the training set into labeled and unlabeled classes. Further, in order to train models on point clouds only, where only the distance to the nearest point in the point set is available, it's necessary to use a loss that is a proxy for SDFs. The proposed loss is a straightforward and improved proxy between nearest point unsigned distance and SDF values relative to prior methods, and includes an additional term to encourage precision closer to the 0-level set.
2. _High performance relative to baselines_: The performance of GenSDF vs. the closest baseline in terms of chamfer distance is 0.351 vs 1.348 for seen and 0.407 vs 2.333 for unseen. This further highlights the poor generalization ability of prior methods.

### Weaknesses
1. _Evaluation to unseen classes_: The Acronym dataset is based on ShapeNetSem. While ShapeNetSem does contain 270 different categories, a lot of these are distinct but semantically similar. For example, Chair, OfficeChair, Stool, SideChair, Barstool, AccentChair are all similar chair-like objects that are distinct categories in ShapeNetSem, and Table, RoundTable, EndTable, DiningTable, CoffeeTable, AccentTable are also all similar table-like objects that are distinct categories in ShapeNetSem. If the dataset contains categories that are distinct, but very similar according to their shape, this may potentially affect to what extent claims can be made about generalization.
2. _Further evaluation on YCB_: The paper shows two qualitative examples on YCB. It would be very helpful to see multiple instances of real world generalization capability, especially of different types of objects. The data is not re-scaled to match the distribution of the original training data. While this makes sense to showcase generalization ability, not doing this pre-processing and showing the results removes a valuable evaluation of the model.

## Clarity
The paper is very well written and easy to follow. The visualizations of the reconstructions and the diagrams are high quality. The tables are clearly formatted.

## Significance
### Strengths
It is useful to develop object surface estimation methods that generalize, so that they can be used in cases where lots of ground truth meshes (necessary to derive SDF for supervision) are not available.

### Weaknesses
Lack of provided context for real world applicability. There is a multitude of ways in which 3D point clouds can be obtained for which we would like to get a detailed and high quality mesh estimate. However, the characteristics, such as the amount and kinds of noise,  or number of points of the point clouds depend on the method they were obtained (e.g. multiple depth sensor readings vs. multi-view stereo). It would be very useful if the paper included some text about how the tasks designed and used for evaluation (e.g. 5K points with a specific kind of Gaussian noise) in the paper are a good proxy for the real world utility of models like GenSDF.

---

> ### Author Response · Authors · 2022-08-02
> **Response to Reviewer 8obm**
>
> > "If the dataset contains categories that are distinct, but very similar according to their shape, this may potentially affect to what extent claims can be made about generalization"
>
> We agree that some categories in the train and test sets have similar shape, and that is easier to reconstruct these "unseen" but similar categories. However, Fig. 3 in the main text reports the **log-scale** per-category histogram of reconstruction quality, and we observe improvements to the entire distribution of Chamfer Distance scores over 100+ categories as compared to existing methods. Qualitatively, in Fig. 5 of the main text and supplementary Figs. 2 and 3, even "outlier" objects such as the thin-structure flamingo, helicopter, and flat-faced scissors are accurately reconstructed despite the fact they do not structurally resemble any category of the train set.
>
> > “It would be very helpful to see multiple instances of real world generalization capability, especially of different types of objects. The data is not re-scaled to match the distribution of the original training data. While this makes sense to showcase generalization ability, not doing this pre-processing and showing the results removes a valuable evaluation of the model.”
>
> While leaving the data un-scaled highlights the generalizability of the method, we agree that pre-processing input point clouds is beneficial. We document results in *Fig. 7 of the revised submission*. After re-centering and normalization, we find our reconstructions further improve and all margins compared to baseline methods are preserved or increased.
>
> > “Lack of provided context for real world applicability. There is a multitude of ways in which 3D point clouds can be obtained for which we would like to get a detailed and high quality mesh estimate. However, the characteristics, such as the amount and kinds of noise, or number of points of the point clouds depend on the method they were obtained (e.g. multiple depth sensor readings vs. multi-view stereo). It would be very useful if the paper included some text about how the tasks designed and used for evaluation (e.g. 5K points with a specific kind of Gaussian noise) in the paper are a good proxy for the real world utility of models like GenSDF.”
>
> We include results on the YCB dataset which is a real-world point cloud dataset acquired from multi-view RGBD captures. The fused multi-view point clouds in this dataset resemble input measurements for a robotic part-picking or manipulation task. We demonstrate robust mesh reconstructions of the measured data, e.g., recovering the "handle" of a pitcher in *Fig. 7 in the revised manuscript*, which may serve as input to complex robotic grasping tasks. In the next version, we will add further discussion of the tasks our work may support and what level of noise in the point cloud it is susceptible to.
>
> To illustrate how our model can adapt to noise that drastically exceeds measurement noise in YCB, we gradually add Gaussian noise with mean zero and variance $\sigma^2$  to the input point cloud and evaluate the CD (* 10-4).
>
> | $\sigma^2$  | 0.0   | 0.01 | 0.05 | 0.1 | 0.15 | 0.2 |
> | --------        | --------        | --------      |------|------|------|------|
> | CD| 0.870 |0.911 | 3.122 | 5.208|7.429|7.391|
>
> Qualitatively, from $\sigma^2 > 0.1$, the reconstructed object degrades severely. Future work could investigate specifically learning noise-robust geometric priors for mesh reconstruction from low-quality point clouds.

---

> > ### Author Response · Authors · 2022-08-07
> > **Follow-up on Rebuttal**
> >
> > Dear Reviewer 8obm,
> >
> > Following your questions, we provided additional explanations and YCB visualizations to illustrate the effectiveness of our approach in real-world settings. In light of this, we would like to know whether you believe we have addressed your concerns, and if so we hope that you would be willing to increase your score.
> >
> > Thank you for your time,
> >
> > The Authors

---

> > > ### Comment · Reviewer_8obm · 2022-08-08
> > > **Response to follow up**
> > >
> > > Thank you for the detailed response.
> > >
> > > Regarding the question about seen and unseen categories being similar, I am satisfied with the description about the CD histogram improving overall. Please make sure to include this discussion. I appreciate the additional results with normalized point clouds, and think they are a valuable addition. I do not have further follow up question.

---

### Author Response · Authors · 2022-08-02
**General Response to All Reviewers**

We thank all reviewers for their thoughtful feedback and we are happy to see the positive reception. We address each reviewer's questions individually below. Additionally, *we have added an Appendix in the revised submission of the main manuscript with additional experimental results such as training curves and figures that we could not attach in the comments*. All changes compared to the previous submission are highlighted in blue color. We reference those results in some of our responses to the individual reviewers.

---

### Meta-Review · Area_Chair_Kjk4 · 2022-08-21

**Recommendation:** Accept
**Confidence:** Certain

**Metareview:**

This paper studies the generalization ability of neural signed distance functions by proposing a two-stage semi-supervised meta-learning framework. The method has been tested on both synthetic data and real point clouds. The paper received a total of 4 reviews.  After the rebuttal, Reviewers 84Fy (accept), 8obm (week accept), GFH2 (week accept) voted for accepting the paper because they reached an agreement that the paper proposes a novel, simple yet effective method to learn generalizable signed distance functions. Reviewer 39TN voted for a “Borderline reject” due to his/her concern about the lack of theoretical understanding of the proposed method, but in the rebuttal, the authors actively replied to Reviewer 39TN’s concern and provided additional intuition and experiments to illustrate the effectiveness of our approach. After an internal discussion, AC recommends accepting the paper because it presents a very useful tool for 3D representation and all the major concerns raised by the reviewers have been addressed during the rebuttal. AC urges the authors to improve their paper by taking into account all the suggestions from reviewers.



**Award:**

No

---

### Decision · Program_Chairs · 2022-09-14

Accept